# Multi-omic machine learning predictor of breast cancer therapy response

Stephen-John Sammut[1,2,3], Mireia Crispin-Ortuzar[1,15], Suet-Feung Chin[1,15], Elena Provenzano[3], Helen A. Bardwell[1], Wenxin Ma[4], Wei Cope[1], Ali Dariush[1,5], Sarah-Jane Dawson[6,7], Jean E. Abraham[2,3], Janet Dunn[8], Louise Hiller[8], Jeremy Thomas[9,10], David A. Cameron[9], John M. S. Bartlett[9,11,12], Larry Hayward[9], Paul D. Pharoah[3,13], Florian Markowetz[1], Oscar M. Rueda[1,14], Helena M. Earl[2,3] & Carlos Caldas[1,2,3]✉

Breast cancers are complex ecosystems of malignant cells and the tumour microenvironment[1]. The composition of these tumour ecosystems and interactions within them contribute to responses to cytotoxic therapy[2]. Efforts to build response predictors have not incorporated this knowledge. We collected clinical, digital pathology, genomic and transcriptomic profiles of pre-treatment biopsies of breast tumours from 168 patients treated with chemotherapy with or without HER2 (encoded by *ERBB2*)-targeted therapy before surgery. Pathology end points (complete response or residual disease) at surgery[3] were then correlated with multi-omic features in these diagnostic biopsies. Here we show that response to treatment is modulated by the pre-treated tumour ecosystem, and its multi-omics landscape can be integrated in predictive models using machine learning. The degree of residual disease following therapy is monotonically associated with pre-therapy features, including tumour mutational and copy number landscapes, tumour proliferation, immune infiltration and T cell dysfunction and exclusion. Combining these features into a multi-omic machine learning model predicted a pathological complete response in an external validation cohort (75 patients) with an area under the curve of 0.87. In conclusion, response to therapy is determined by the baseline characteristics of the totality of the tumour ecosystem captured through data integration and machine learning. This approach could be used to develop predictors for other cancers.

Neoadjuvant treatment, that is, systemic therapy (chemotherapy with or without targeted therapy) administered before surgery, is increasingly used in the management of breast cancer to improve rates of breast-conserving surgery and increase survival[4]. However, many patients do not have a good response[3,5]. Features associated with response to neoadjuvant therapy have been derived from clinical[6], molecular[7–12] and digital pathology analysis[13,14]. However, these studies have been frequently small, combined data from patients receiving different treatments and used single platform profiling that fails to capture the complexity of the tumour ecosystem. Unsurprisingly, physicians continue to select patients for neoadjuvant therapies using empirical clinical risk-stratification[15].

Tumour ecosystems are increasingly recognized as major determinants of treatment response[2] and we hypothesized that improved prediction models need to account for tumours as complex ecosystems, comprising communities of malignant clones within a microenvironment of stromal, vascular and immune cell types that are perturbed by therapy.

Here we characterized biological parameters extracted from a prospective neoadjuvant study that collected detailed pre-therapy tumour multi-omic data and associated these with eventual response. We found that malignant cell, immune activation and evasion features were associated with treatment response. These features, derived from clinicopathological variables, digital pathology and DNA and RNA sequencing, were used as input into an ensemble machine learning approach to generate predictive models. We validated the accuracy of the predictive models in independent, external cohorts and demonstrated that the best performers integrated clinicopathological and molecular data. The overall approach is widely applicable to other cancers and can be customized to include both fewer and newer features.

## Multi-platform profiling of tumour biopsies

We prospectively enrolled 180 women with early and locally advanced breast cancer undergoing neoadjuvant treatment into a molecular

[1]Cancer Research UK Cambridge Institute, University of Cambridge, Li Ka Shing Centre, Cambridge, UK. [2]Department of Oncology, University of Cambridge, Cambridge, UK. [3]CRUK Cambridge Centre, Cambridge Experimental Cancer Medicine Centre (ECMC) and NIHR Cambridge Biomedical Research Centre, University of Cambridge and Cambridge University Hospitals NHS Foundation Trust, Cambridge, UK. [4] School of Clinical Medicine, University of Cambridge, Cambridge, UK. [5]Institute of Astronomy, University of Cambridge, Cambridge, UK. [6]Peter MacCallum Cancer Centre, Melbourne, Victoria, Australia. [7]Centre of Cancer Research and Sir Peter MacCallum Department of Oncology, University of Melbourne, Melbourne, Victoria, Australia. [8]Warwick Clinical Trials Unit, University of Warwick, Coventry, UK. [9]Edinburgh Cancer Research Centre, Western General Hospital, Edinburgh, UK. [10]Q2 Laboratory Solutions, Livingston, UK. [11]Ontario Institute for Cancer Research, Toronto, Ontario, Canada. [12]Laboratory Medicine and Pathobiology, University of Toronto, Toronto, Ontario, Canada. [13] Strangeways Research Laboratory, University of Cambridge, Cambridge, UK. [14]MRC Biostatistics Unit, University of Cambridge, Cambridge, UK. [15]These authors contributed equally: Mireia Crispin-Ortuzar, Suet-Feung Chin. ✉e-mail: carlos.caldas@cruk.cam.ac.uk

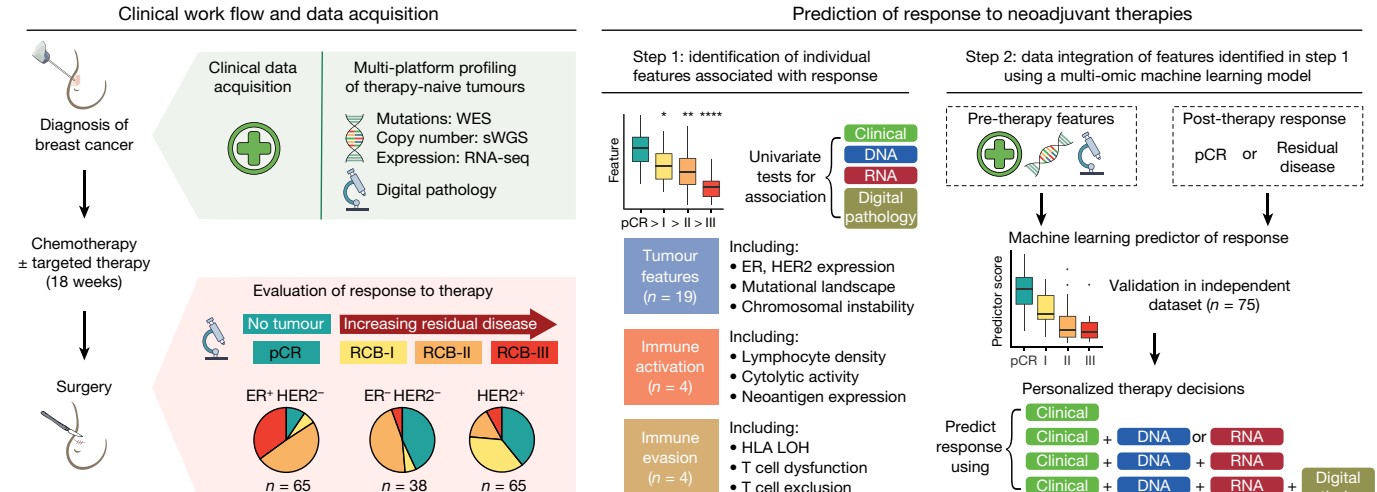

**Fig. 1 | Overview of the study design.** Pre-therapy breast tumours from 168 patients were profiled using DNA sequencing and RNA sequencing (RNA-seq) and digital pathology analysis. Response was assessed on completion of neoadjuvant therapy using the RCB classification. Individual pre-therapy clinical, molecular and digital pathology features associated with pCR were identified and integrated within machine learning models to predict responses, which were then validated in an independent dataset. sWGS, shallow whole-genome sequencing; WES, whole-exome sequencing.

profiling study (TransNEO) (Fig. 1, the cohort characteristics are summarized in Supplementary Table 1). Fresh-frozen pre-treatment core tumour biopsies were collected from 168 cases using ultrasound guidance (Extended Data Fig. 1). DNA and RNA were extracted and profiled by shallow whole-genome sequencing (168 samples), whole-exome sequencing (168 samples) and RNA sequencing (162 cases). The diagnostic core biopsy haematoxylin and eosin-stained slides from 166 cases were digitized. The tumours sampled (*n* = 168) included all major subtypes of breast cancer. Chemotherapy (block-sequential taxane and anthracycline) was administered for a median of 18 weeks (6 cycles) in 145 cases; 22 cases received a taxane (in combination with carboplatin in 3 cases and cyclophosphamide in 13 cases) and 1 case received an anthracycline in combination with cyclophosphamide. Two patients received only one cycle owing to drug toxicities (Supplementary Table 1). Patients with HER2+ tumours (*n* = 65) received a median of three cycles of anti-HER2 therapy in combination with a taxane. Response was assessed at surgery using the residual cancer burden (RCB) classification[3,5] (Extended Data Fig. 2a, b). On completion of neoadjuvant treatment, in the 161 cases with RCB assessment, 42 (26%) had a pathological complete response (pCR), 25 (16%) had a good response (RCB-I), 65 (40%) had a moderate response (RCB-II) and 29 (18%) had extensive residual disease (RCB-III).

## Clinical phenotypes are limited predictors

The clinical features individually associated with pCR (Extended Data Fig. 2c, d; univariable logistic regression) included tumour grade (odds ratio (OR): 4.2, confidence interval (CI): 1.8–11, false discovery rate (FDR) = 0.009), ER− receptor status (OR: 4.2, CI: 2–9.1, FDR = 0.002) and absence of lymph node involvement at diagnosis (OR: 3, CI: 1.4–6.6, FDR = 0.01). When all of these variables were combined by multiple logistic regression, only ER− status was associated with pCR (OR: 3.8, CI: 1.6–9.2, FDR = 0.009), but there was response heterogeneity (for example, 55% of ER− tumours did not attain pCR).

## Genomic landscapes associate with response

Whole-exome sequencing (*n* = 168 tumours) identified 16,134 somatic mutations (Supplementary Table 2), with the highest frequency in driver genes, including *TP53* (*n* = 96, 57%), *PIK3CA* (*n* = 44, 26%), *GATA3* (*n* = 16, 10%) and *MAP3K1* (*n* = 13, 8%) (Extended Data Figs. 3, 4a). *TP53* mutations were associated with pCR (OR: 2.9, CI: 1.3–6.6, *P* = 0.01;

Extended Data Fig. 4a), as previously reported[7], whereas *PIK3CA* mutations were associated with residual disease (OR: 2.1, CI: 1.3–3.4, *P* = 0.002).

Tumour mutation burden was higher in tumours that attained pCR (median mutations per megabase pCR: 2.3, residual disease: 1.4, *P* = 0.0005) and monotonically associated with RCB class (*P* = 0.004; Fig. 2a). This was independent of computationally estimated tumour purity (Extended Data Fig. 4b). In subgroup analysis, the association was observed only in HER2− (*P* = 9 × 10⁻⁶) tumours (Extended Data Fig. 4c). The clonal status of mutations[16] also associated with response: tumours that failed to attain pCR had a higher percentage of subclonal mutations (Fig. 2b). Accordingly, tumours that attained pCR had higher predicted neoantigen burdens (median neoantigens pCR: 28, residual disease: 17, *P* = 0.009; Fig. 2c), and after stratification, this was observed only in HER2− tumours (*P* = 0.004; Extended Data Fig. 4d).

Analysis of mutational signatures[17] (Fig. 2d) showed homologous recombination deficiency (HRD) and APOBEC signatures were associated with pCR across the entire cohort (HRD OR: 1.1, *P* = 0.006; APOBEC OR: 1.1, *P* = 0.02; logistic regression). Tumours that attained pCR had a greater contribution from non-clock signatures (*P* = 0.002; Extended Data Fig. 4e). Similarly, increasing HRD[18] was monotonically associated with response (*P* = 0.00001; Fig. 2e) and associated with pCR in HER2− tumours (*P* = 3 × 10⁻⁶; Extended Data Fig. 4f).

Tumours that attained pCR had more copy number alterations and chromosomal instability was monotonically associated with RCB class (*P* = 0.0002; Fig. 2f, Extended Data Fig. 4g). To capture the ensemble of copy number alterations, which dominate the genomic landscapes, we stratified the pre-treated tumours into the 10 genomic driver-based integrative cluster (iC) subtypes[19] (Extended Data Fig. 4h). iC10 tumours, mostly triple-negative with high prevalence of *TP53* mutations and copy number alterations, showed the strongest association with pCR. By contrast, tumours from indolent ER+ subtypes, iC3, iC7 and iC8 were unlikely to attain pCR. Two of the aggressive ER+ subtypes, iC2 (11q13/14 amplification) and iC6 (amplification of *ZNF703* at 8p12), also associated with lack of treatment response. We had previously reported a similar association for iC2 tumours[20].

In summary, tumours that attained pCR mostly came from more-aggressive iC subtypes, were enriched for *TP53* mutations, had higher tumour mutation burdens and neoantigen loads, had less-complex clonal architectures and were enriched for APOBEC and HRD signatures.

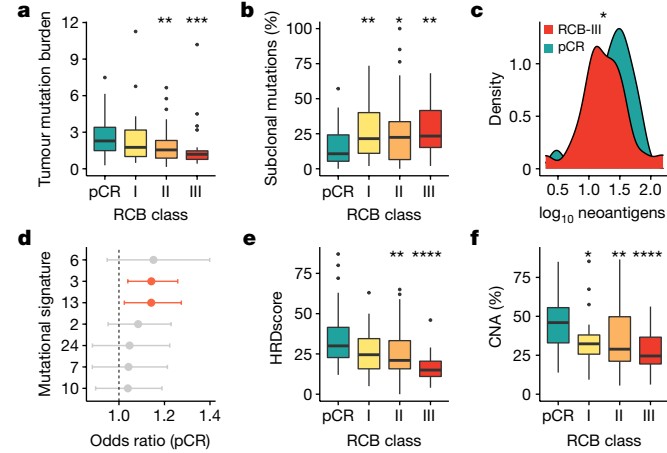

**Fig. 2 | Genomic features monotonically associate with response to therapy. a, b**, Box plots showing monotonic association between RCB class: total tumour mutation burden (**a**) ($P = 0.004$, ordinal logistic regression; pCR versus RCB-II **$P = 0.001$ and RCB-III ***$P = 0.0002$), and the percentage of subclonal mutations (**b**) ($P = 0.02$, ordinal logistic regression; pCR versus RCB-I **$P = 0.007$, RCB-II *$P = 0.04$ and RCB-III **$P = 0.001$). **c**, Density curves showing distribution of neoantigens in tumours that attained pCR and RCB-III (monotonic association, $P = 0.03$, ordinal logistic regression; pCR versus RCB-III *$P < 0.05$). **d**, Associations between mutational signatures and pCR. Statistically significant associations obtained from logistic regression are shown in red (HRD: 3; APOBEC: 13). The measure of the centre is the parameter estimate, and the error bars represent 95% confidence intervals; the vertical dashed line corresponds to an odds ratio of 1. **e, f**, Box plots showing monotonic association between RCB class and HRD score (**e**) ($P = 0.00001$, ordinal logistic regression; pCR versus RCB-II **$P = 0.006$ and RCB-III ****$P = 3 \times 10^{-6}$), and the percentage of copy number alterations (CNAs; **f**) ($P = 0.0002$, ordinal logistic regression; pCR versus RCB-I *$P = 0.01$, RCB-II **$P = 0.004$ and RCB-III ****$P = 7 \times 10^{-5}$). In **a–f**, the number of patients with DNA sequencing data: 40 (for pCR), 24 (for RCB-I), 64 (for RCB-II) and 27 (for RCB-III). In **a, b, e, f**, the box bounds the interquartile range divided by the median, with the whiskers extending to a maximum of 1.5 times the interquartile range beyond the box. Outliers are shown as dots. Wilcoxon rank-sum tests; all $P$ values are two-sided.

## HLA class I allelic loss confers resistance

Loss of heterozygosity (LOH) over the HLA class I locus[21] was identified in 29 cases and associated with residual disease (OR: 3.5, CI: 1.1–14.2, $P < 0.05$; logistic regression) independently of global LOH and copy number instability (Extended Data Fig. 4i). HLA LOH events were predicted to result in inability to present 30% of tumour neoantigens and 69% of LOH events targeted HLA alleles that presented an equal or greater number of neoepitopes than the retained allele. These data support a model in which some tumours appear to have immune escaped by losing copies of the HLA locus and these tumours are less likely to respond to treatment.

## Tumour proliferation and immune signatures

We modelled response as a binary variable (pCR versus residual disease) and differential RNA expression analysis showed 2,071 genes underexpressed and 2,439 genes overexpressed in tumours attaining pCR (FDR < 0.05). pCR associated with overexpression of driver genes *CDKN2A*, *EGFR*, *CCNE1* and *MYC* and underexpression of *CCND1* (iC2), *ZNF703* (iC6) and *ESR1* (Fig. 3a). Gene set enrichment analysis on the MsigDB Hallmarks[22] and Reactome[23] gene sets showed that proliferation and immune activation strongly associated with response (Fig. 3b, Extended Data Fig. 5a, b).

To further explore this association, we performed gene set variation analysis using the Genomic Grade Index (GGI) gene set[24] (Supplementary Table 3). The GGI gene set variation analysis score associated with tumour grade (Fig. 3c, left panel) and was monotonically associated

with RCB class ($P = 2 \times 10^{-5}$; Fig. 3c, middle panel). Similar results were observed on enriching over an embryonic stem-cell metagene[25] ($P = 0.0001$; Fig. 3c, right panel), indicating that tumour dedifferentiation associates with response. In a subgroup analysis, this association was only observed in HER2[-] tumours ($P = 4 \times 10^{-5}$; Extended Data Fig. 6a), suggesting that efficacy of anti-HER2-targeted therapies is independent of proliferation. We also explored the utility of a taxane response metagene[26], computed as the difference in expression of proliferation and ceramide metagenes: HER2[-] tumours that attained pCR had higher enrichment scores ($P = 5 \times 10^{-7}$; Extended Data Fig. 6b).

The role of the tumour immune microenvironment (TiME) in predicting response was suggested by the automated scoring of digitally scanned core biopsy haematoxylin and eosin slides showing that lymphocytic density was a good predictor of pCR ($P = 0.0006$; Fig. 3d, left panel), in line with previous reports[13,14]. The immune cytolytic activity score[27] was also monotonically associated with response across all tumours ($P = 0.001$; Fig. 3d, middle panel) and correlated with tumour lymphocytic density ($R^2 = 0.4$, $P = 1 \times 10^{-15}$).

These results motivated a detailed analysis of the TiME in pre-treatment biopsies using three different methods for RNA expression deconvolution (enrichment over Danaher gene sets[28], MCPcounter[29] and Immunophenoscore[30]; Fig. 3d, right panel, Extended Data Fig. 7a–d). These analyses converged to reveal enrichment of both innate and adaptive immunity cell populations in ER[+]HER2[-] and HER2[+] tumours that attained pCR. Computationally estimated lymphocyte density also strongly correlated with the enrichment of many adaptive and innate immune components (Extended Data Fig. 7e). Immunologically active tumours were co-enriched for both cytotoxic and immunoinhibitory cell types and gene signatures (Extended Data Fig. 7d). The Danaher gene set enrichment also showed that mast cells were enriched in resistant tumours (enrichment score pCR: 2.1, residual disease: 3.4, $P = 0.0001$).

We then integrated proliferation (using GGI) and immune response in the pre-therapy tumours. We used the STAT1 gene expression module[31] to represent immune response in a single score and computed correlations between GGI and STAT1 scores with RCB classes. Tumours that attained pCR mostly had high proliferation and high immune activation, with both signatures decreasing in a stepwise manner as the degree of residual disease increased (Fig. 3e). Similar findings were observed on analysing external data from the ISPY-I and NCT00455533 studies[10,11] (Extended Data Fig. 7f).

In summary, in therapy-naive tumours, proliferation and immune response, both innate and adaptive, have combined effects that associate with sensitivity to treatment. In general, tumours that attain pCR tend to be highly proliferative and display evidence of an active TiME.

## Immune dysfunction in resistant tumours

We noted that there were 26 of the 45 tumours with high GGI and STAT1 scores that failed to attain pCR. Differential gene expression analysis in these 45 cases (residual disease versus pCR) showed enrichment of epithelial-to-mesenchymal transition and downregulation of immune response pathways in tumours with residual disease (Fig. 3f). We hypothesized that an attenuated immune response could explain this and derived T cell dysfunction and T cell exclusion metrics using TIDE[32] (Fig. 3f). This showed that HER2[-] tumours with residual disease had higher T cell dysfunction at diagnosis ($P = 0.006$) with no difference in T cell exclusion scores. The increased dysfunction was associated with enrichment of inhibitory natural killer CD56[dim] cells ($P = 0.01$) and regulatory T cells ($P = 0.02$; Extended Data Fig. 8a). Across the whole cohort, active T cell exclusion (Extended Data Fig. 8b) was associated with poorer response: exclusion was higher in residual disease ($P = 0.02$), with increased enrichment of cancer-associated fibroblasts ($P = 0.009$) and M2 tumour-associated macrophages ($P = 0.0009$).

In summary, some tumours, despite being proliferative and with an enriched TiME, display features of T cell dysfunction and tend to be resistant to therapy.

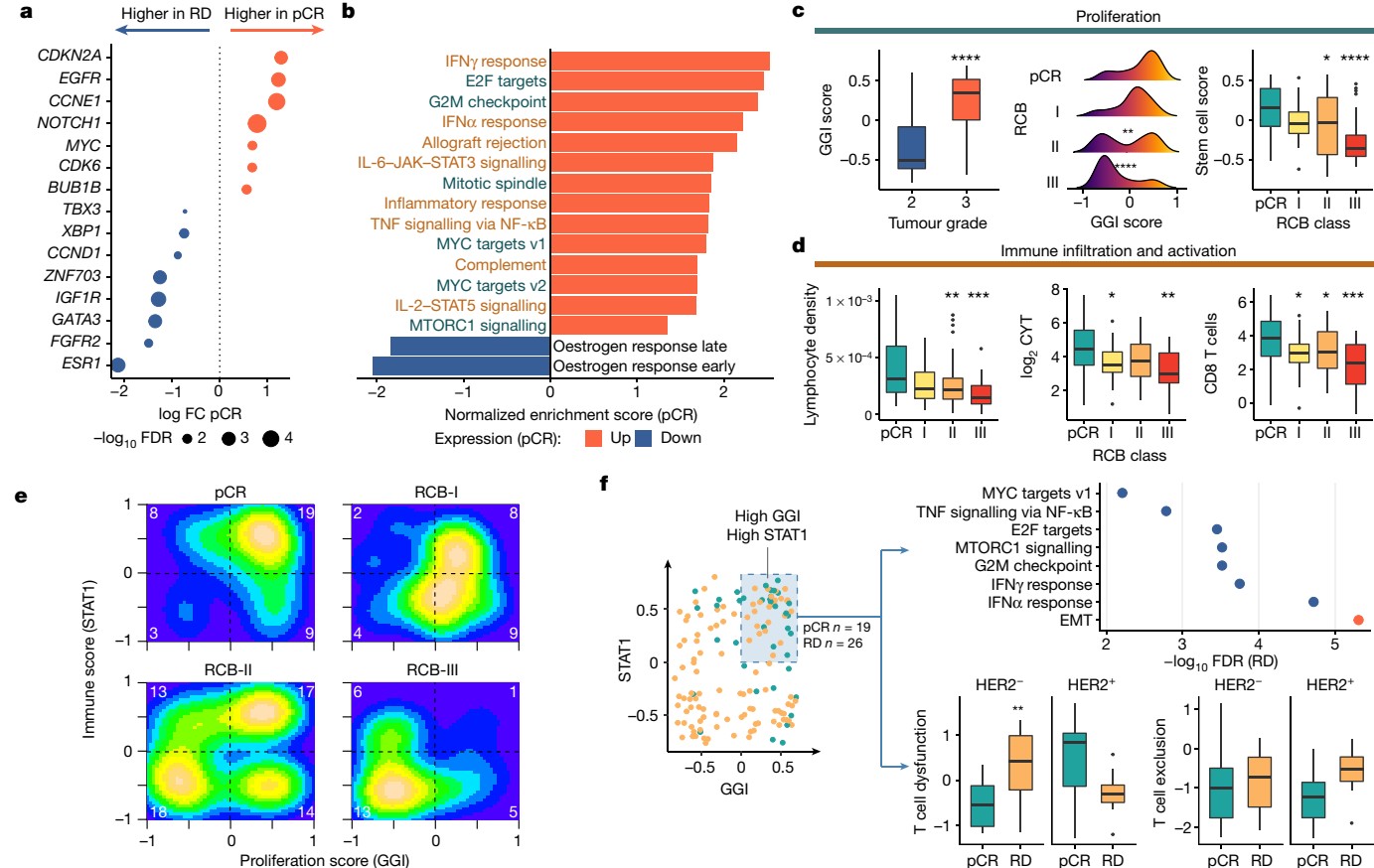

**Fig. 3 | Transcriptomic features associated with response to neoadjuvant therapy. a**, Expression of breast cancer driver genes associated with pCR. FC, fold change; RD, residual disease. **b**, MSigDB Hallmark gene sets associated with pCR. Response was predominantly associated with proliferative (green) and immune (brown) gene sets. **c**, Box plot showing association of GGI score with histological grade ($P = 5 \times 10^{-11}$) (left); density plots showing monotonic association ($P = 2 \times 10^{-5}$, ordinal logistic regression) between GGI score and RCB (pCR versus RCB-II **$P = 0.01$ and RCB-III ****$P = 3 \times 10^{-5}$) (middle); and box plot showing monotonic association ($P = 0.0001$, ordinal logistic regression) between stem-cell enrichment score and RCB (pCR versus RCB-II *$P = 0.02$ and RCB-III ****$P = 7 \times 10^{-5}$) (right). The number of patients with RNA sequencing data: 39 (for pCR), 23 (for RCB-I), 62 (for RCB-II) and 25 (for RCB-III). **d**, Box plots showing monotonic associations between computationally estimated lymphocyte density and RCB ($P < 1 \times 10^{-10}$, ordinal logistic regression; $n = 153$ cases with digital pathology data; pCR versus RCB-II **$P = 0.006$ and RCB-III ***$P = 0.0001$) (left); CYT score and RCB ($P = 0.001$; $n = 149$ cases with RNA

sequencing data; pCR versus RCB-I *$P = 0.03$ and RCB-III **$P = 0.001$) (middle); and Danaher CD8 T cell enrichment and RCB ($P = 0.0002$; $n = 149$ cases; pCR versus RCB-I *$P = 0.04$, RCB-II *$P = 0.04$ and RCB-III ***$P = 0.0003$) (right). **e**, 2D density plot showing the relationship between proliferation and immune activation across RCB classes. The number of cases in each quadrant is shown in white. **f**, The distribution of GGI and STAT1 scores across cohort (left). The shaded area represents samples with proliferation and immune enrichment values above the mean ($n = 45$ cases). The MSigDB Hallmarks pathways associated with residual disease in these 45 tumours (red represents overexpressed, and blue indicates underexpressed) (top right). Box plots showing association between T cell dysfunction (**$P = 0.006$ HER2$^-$) and exclusion with response in these tumours are also shown (bottom right). EMT, epithelial-to-mesenchymal transition. In **c**, **d**, **f**, the box bounds the interquartile range divided by the median, with the whiskers extending to a maximum of 1.5 times the interquartile range beyond the box. Wilcoxon rank-sum tests; all $P$ values are two-sided.

## Machine learning integrates multi-omic features

Above, we identified clinical, digital pathology, genomic and transcriptomic features present in the naive tumour ecosystem that associated with response to therapy, although individually none of these features performed robustly. This motivated the use of a machine learning framework (Fig. 4a) to integrate features into a predictive model of pCR.

A series of six pCR prediction models including different feature combinations were derived using: (1) clinical features only, and adding (2) DNA, (3) RNA, (4) DNA and RNA, (5) DNA, RNA and digital pathology, and (6) DNA, RNA, digital pathology and treatment. The number of predictive features totalled 34 (Fig. 4b, Extended Data Fig. 9a, b, Supplementary Table 4).

The models were based on a multi-step predictor pipeline. Inside the pipeline, features were first filtered by univariable selection and collinearity reduction, and then fed into an unweighted ensemble classifier[33]. Each ensemble consisted of three algorithms acting in parallel: logistic regression with elastic net regularization, a support vector machine

and a random forest. The three algorithm scores were then averaged to form the predictor (Extended Data Fig. 9c). A fivefold cross-validation scheme was used to optimize model hyperparameters (Methods and Supplementary Methods).

The fully trained models were tested for validation on an independent external cohort of 75 patients that received neoadjuvant therapy, either cases randomized to the control arm of the ARTemis clinical trial[34] or cases recruited into the Personalised Breast Cancer Programme (details listed in Supplementary Table 5). In the external cohort, the models achieved the following areas under the curve: 0.70 (clinical), 0.80 (clinical and DNA), 0.86 (clinical and RNA), 0.86 (clinical, DNA and RNA), 0.85 (clinical, DNA, RNA and digital pathology), 0.87 (fully integrated model (clinical, DNA, RNA, digital pathology and treatment)) (Fig. 4c, d, Extended Data Fig. 9d, e). The baseline clinical model, as implemented using our machine learning algorithms, performed similarly to other clinical predictors reported in larger datasets[35,36].

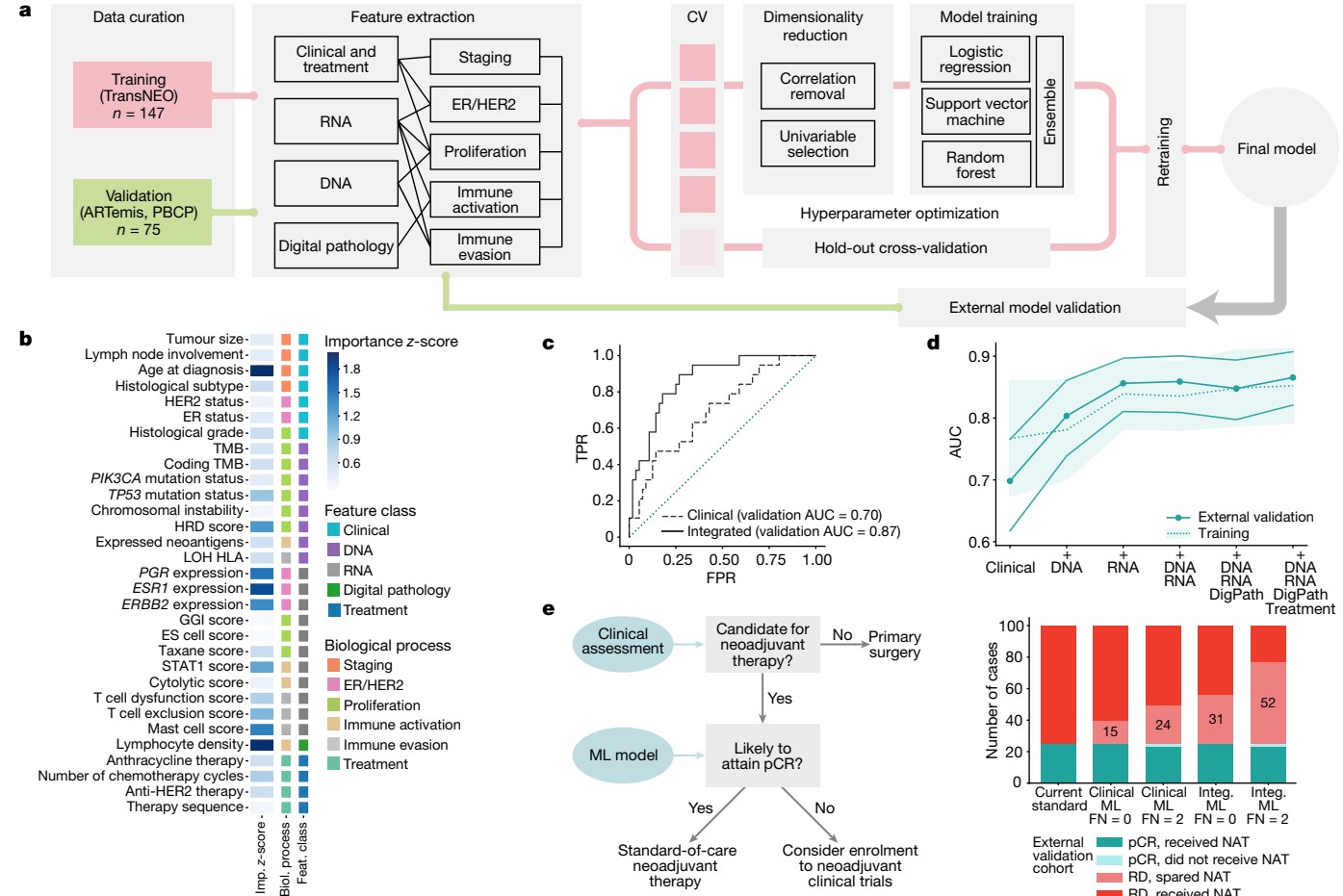

**Fig. 4 | Predicting response to therapy using a composite machine learning model. a**, Schematic of the machine learning framework. CV, cross validation. **b**, Feature importance calculated as the average z-score resulting from dropping each individual feature from the three components of the model and calculating the new area under the receiver operating characteristic curve (AUC). The importance of chemotherapy sequence features have been averaged into a 'therapy sequence' row for simplicity. ES cell, embryonic stem cell; TMB, tumour mutation burden. **c**, Receiver operating characteristic curves for the clinical (dashed) and fully integrated (continuous) models applied on the external validation cohort. The dotted line indicates random performance. FPR, false positive rate; TPR, true positive rate. **d**, AUCs for models with increasing levels of data integration. The continuous line on the foreground corresponds to the AUCs obtained from the external validation cohorts (filled markers), with bands representing the standard deviation estimated with bootstrap. The filled band on the background corresponds to the standard deviation of the AUCs obtained using cross-validation on the training dataset, with mean values represented by a dashed line. DigPath, digital pathology. **e**, Potential clinical impact of the pCR model, using data from the external validation confusion matrix (left). Bar plots show the number of patients that would be identified to be chemoresistant using operating thresholds of 0 and 2 false negatives (FN), using either the clinical or fully integrated models, respectively (right). ML, machine learning; NAT, neoadjuvant therapy.

We explored the importance of the features used in the integrated training model and found that it used clinical phenotypes in combination with DNA, RNA and digital pathology features. The dominant features were age, lymphocyte density, and expression of *PGR*, *ESR1* and *ERBB2* (Fig. 4b, Extended Data Fig. 9b, Supplementary Table 6). In addition, the predictive model also used features associated with proliferation, immune activation and immune evasion. The fully integrated model relied on features obtained from all modalities of data, with RNA features having the largest contribution (Fig. 4b, Extended Data Fig. 9b).

Despite the models being trained using a binary response variable (pCR versus residual disease), an analysis of the predictor scores across both training and validation sets showed that these were highly correlated with RCB class, with a monotonic association observed (training: $P = 3 \times 10^{-10}$, validation: $P = 1 \times 10^{-6}$; Extended Data Fig. 10).

In a clinical workflow, the predictive models could be applied to candidates for neoadjuvant therapy; any predicted to have chemoresistant tumours should be considered for enrolment into clinical trials of novel therapies, as their prognosis is poor if they are treated with standard-of-care therapies (Fig. 4e). We explored this in a simulation study and applied the confusion matrix obtained in the external validation cohorts on a total of 100 patients about to receive neoadjuvant therapy. If the criterion was that no patient guaranteed to obtain pCR should miss out on treatment (no false negatives), the clinical machine learning model would identify 15 non-responders, whereas the fully integrated machine learning model would increase this number to 31. By relaxing the false-negative threshold and allowing two false negatives, 24 (clinical model) and 52 (fully integrated model) patients who would not attain pCR would be correctly identified (Fig. 4e).

In summary, we used an ensemble machine learning approach that inputs multi-omic features from the pre-treatment biopsy to derive predictors of pCR. The models were externally validated demonstrating very good discrimination power.

## Discussion

Human tumours are complex ecosystems formed in the malignant compartment by communities of clones and cell phenotypes, and in the tumour microenvironment by a very diverse array of stromal,

vascular, innate and adaptive immune cell types[1,2,37]. How these ecosystems are organized in breast cancer appears to be strongly associated with their genomic features[38]. Therapy perturbs these tumour ecosystems and this is increasingly recognized as one of the main determinants of treatment response[2]. Remarkably, efforts to identify features in untreated tumours that predict response to therapy have mostly ignored this.

Our findings showed that response is determined to a great degree by the baseline characteristics of the totality of the tumour ecosystem. Tumour proliferation emerged as a key determinant of response as reported previously[9,26]. Genomic features that associated with response to chemotherapy in HER2⁻ tumours, and usually correlated with proliferation, included *TP53* mutations, tumour mutation burden, BRCA, HRD and APOBEC mutational signatures, and chromosomal instability. Remarkably, in HER2⁺ tumours, treated with chemotherapy and HER2-targeted antibodies, response appeared to be independent of proliferation. This observation was previously reported[39] and should motivate a search for the underlying mechanism. Clonal diversity and subclonal mutations were associated with residual disease. This has also been reported in oesophageal carcinoma[40], suggesting that clonally diverse tumours are more likely to contain or be able to select resistant subclones.

A central finding was that the TiME in treatment-naive tumours is a major determinant of response to therapy. Previous work in mouse models had shown that an effective response to chemotherapy requires an immunocompetent tumour microenvironment[41]. Deconvolution of immune subpopulations using our RNA expression data suggested that both innate and adaptive immunity were already engaged in tumours that went on to have pCR. We previously reported digital pathology-derived lymphocytic density as an independent predictor of pCR[13,14], and here confirm this and also show that it strongly correlates with the cytolytic activity score (a surrogate for CD8 and natural killer cytotoxic cells). Pathologist-assessed infiltration of tumour lymphocytes has been reported by many groups as a predictor of response to chemotherapy[42] and immunotherapy[43], and international guidelines for scoring exist[44]. The direct role of the immune system in killing tumour cells as a result of chemotherapy, so-called chemotherapy-induced immunogenic cell death, has been proposed[45]. We hypothesize that the presence of an engaged immune infiltrate in the tumour microenvironment in therapy-naive tumours mediates such chemotherapy-induced immunogenic cell death.

By contrast, a suppressed immune response in naive tumours associated with a propensity for poor response. HLA LOH was first implicated in immune evasion in lung cancer[21] and we show here that it predicts poor response to therapy. T cell dysfunction[46] and exclusion[47] showed similar effects. The similarity of features predicting response to cytotoxic therapy compared with those reported to predict response to immune checkpoint inhibitors[48] raises the intriguing possibility that similar mechanisms of killing tumour cells are engaged.

We show that machine learning models for prediction of therapy response that combine clinical, molecular and digital pathology data significantly outperform those based on clinical variables. The high accuracy obtained in external validation suggests that the models are robust and may enable using molecular and digital pathology to determine therapy choice in future clinical trials, including in the adjuvant therapy setting. More generally, the framework highlights the importance of data integration in machine learning models for response prediction and could be used to generate similar predictors for other cancers.

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

## Methods

### Study population and tissue collection

We analysed breast tumours from patients with primary invasive cancer enrolled in the TransNEO study at Cambridge University Hospitals NHS Foundation Trust between 2013 and 2017. Appropriate ethical approval from the institutional review board (research ethics reference: 12/EE/0484) was obtained for the use of biospecimens with linked pseudo-anonymized clinical data. All patients provided informed consent for sample collection and all participants consented to the publication of research results. Clinical data were collected in Microsoft Excel (as part of the office 365 suite) by data managers.

Pre-neoadjuvant and post-neoadjuvant chemotherapy specimens were handled following departmental standard operating procedures in accordance with international guidelines[49]. RCB post-neoadjuvant therapy was assessed by experienced breast histopathologists (E.P. and J.T.) using the pathology protocol for assessment of RCB as provided on the MD Anderson RCB website (https://www.mdanderson.org/education-and-research/resources-for-professionals/clinical-tools-and-resources/clinical-calculators/calculators-rcb-pathology-protocol2.pdf?_ga=2.93785373.1680005878.1594213442-1702172112.1568299785). RCB assessment was not available in seven cases (Extended Data Fig. 1). pCR was defined as the absence of residual invasive cancer on haematoxylin and eosin (H&E) evaluation of the complete resected breast specimen and all sampled lymph nodes. Results for oestrogen receptor (ER) and HER2 status were extracted from pathology reports. ER and HER2 testing were performed in an accredited diagnostic laboratory and scored according to UK guidelines[50]. ER staining was regarded as positive if the Allred score was more than 2. HER2 was regarded as positive if immunohistochemical staining was 3+, or if there was borderline 2+ staining with *HER2* gene amplification on FISH (HER2 copy number ≥ 6.0 and/or HER2:CEP17 ratio ≥ 2).

Whole blood from all patients was collected before commencing neoadjuvant therapy in S-Monovette 7.5 ml EDTA tubes and centrifuged at 820$g$ for 10 min at room temperature to partition plasma, buffy coat and erythrocytes. The buffy coat fraction was isolated and suspended in 10 ml of red cell lysis buffer (155 mM $NH_4Cl$, 10 mM $KHCO_3$ and 0.1 mM EDTA pH 7.4), centrifuged at 3,600$g$ for 10 min at room temperature, followed by a further step of resuspension and centrifugation. The final cell pellet was suspended in 1 ml of phosphate-buffered saline, centrifuged at 10,000 r.p.m. for 5 min, isolated and frozen. Tumour tissue was collected before the initiation of neoadjuvant chemotherapy via an ultrasound-guided biopsy, flash-frozen in liquid nitrogen and stored at −80 °C. Sectioning of the samples was performed on a cryostat (CM1520; Leica Biosystems). Following an initial 6-μm section taken for H&E staining, 20 30-μm sections were taken and 10 sections were placed in each of the two tubes containing either 180 μl ATL buffer or 700 μl of QIAzol for DNA or RNA extraction, respectively. The histology slides were stained with H&E, and tumour, stromal and immune infiltrate quantification was performed.

### Nucleic acid processing and library preparation

Isolation of DNA from all buffy coat and sectioned tumour tissue samples was performed using the Qiagen DNeasy Blood and Tissue Kit (catalogue no. 69506). DNA from tumour tissue was extracted using the manufacturer-recommended protocol. DNA quantification was performed using the Qubit Fluorometer (Invitrogen) and nucleic acid purity was assessed using the NanoDrop 8000 (Thermo Fisher Scientific). Normal and tumour DNA samples were normalized to a concentration of 5 ng/μl using a fluorescence-based method (Quant-IT dsDNA BR, Q33130, Thermo Fisher Scientific) and 50 ng of DNA used for exome library preparation. DNA libraries were constructed using the Illumina Nextera Rapid Capture Exome Library Preparation kit according to the manufacturer's protocol (Illumina document number:

15037436). The resulting whole-genome sequencing (WGS) libraries and captured whole-exome sequencing (WES) libraries were normalized and pooled, with each pool normalized to a molarity of 4 nM. Sequencing was performed on an Illumina HiSeq4000 instrument in 50-bp single-read mode (for shallow WGS (sWGS)) or 75-bp paired-end mode (for WES). Demultiplexing was performed using Illumina's bcl2fastq2 software using default options. Isolation of RNA from all tumour tissue samples was performed using the Qiagen miRNeasy Mini Kit (catalogue no. 217004). Tissue sections suspended in 700 μl of QIAzol were thawed and mixed by vortexing. Chloroform (140 μl) was added to each sample, vortexed and transferred to a heavy phase lock tube (Qiagen MaXtract, catalogue no. 129056). The samples were then spun at 12,000$g$ for 15 min at 4 °C, following which the upper clear phase containing RNA was transferred to a 2-ml Eppendorf tube. Subsequent extraction was then performed using the Qiagen QIAsymphony using the manufacturer-recommended protocol. RNA quantification was performed using the Qubit Fluorometer (Invitrogen) and assessment of the RNA integrity performed using the high-sensitivity RNA assays on the Agilent 4200 TapeStation Instrument. RNA samples were normalized to a concentration of 10 ng/μl and transcriptomic libraries were prepared using the Illumina TruSeq Stranded mRNA Library Preparation kit (catalogue no. 20020595) according to the manufacturer's protocol (Illumina document number: 1000000040498). Of each library, 5 nM was prepared and 94 samples were pooled per lane of sequencing on an Illumina HiSeq4000 system run in 75-bp paired-end mode. Demultiplexing was performed using bcl2fastq2 v.2.17 software (Illumina) using default options.

### sWGS and WES pre-processing

For each exome paired FASTQ file, sequencing quality metrics were generated using the FastQC tool (version 0.11.7) (https://www.bioinformatics.babraham.ac.uk/projects/fastqc/). Alignment to the GRCh37 decoy assembly of the human genome was performed using Novoalign (version 3.2.13) in paired-end mode with the following parameters enabled: (1) base quality recalibration, (2) trimming of Nextera adaptor sequence CTGTCTCTTATA, and (3) hard clipping of trailing bases with quality ≤ 20. sWGS data were processed similarly; however, Novoalign was run in single-read mode. Binary aligned sequencing (BAM) file merging, coordinate sorting and PCR and optical duplicate marking were performed using Novosort (version 3.2.13). Local realignment around insertions and deletions was performed using the Genome Analysis Toolkit (GATK)[51] programs RealignerTargetCreator and IndelRealigner. The performance of the library preparation as well as the quality of the sequencing data, target coverage metrics within exonic regions specified by the Nextera target BED file obtained from Illumina (Manifest version 1.2) were generated using Picard (version 2.17.0) CalculateHSMetrics. Median WES coverage was ×162 for tumours and ×137 for normal tissue. Median sWGS coverage was ×0.1.

### Variant calling

Germline variants were identified across all tumour and normal samples using GATK HaplotypeCaller (version 4.1.4) run in GVCF mode and filtered using GATK VariantRecalibrator. Somatic variant calling was performed using Mutect2 (version 4.1.4). A panel of normals was created by running Mutect2 in tumour-only mode on all normal samples and the resulting VCF files were merged using CreateSomaticPanelOfNormals. Mutect2 was run on each tumour–normal sample pair using this panel of normals and a database of germline variants present within gnomAD to improve somatic calling. Variant filtration was performed using FilterMutectCalls using default options. Mutations that were present at an allelic fraction (AF) of less than 1%, had coverage of less than ×25 in both normal and tumour tissue exome data, were present in the gnomAD repository with a population prevalence greater than 1% and identified as lying within repetitive regions by ANNOVAR (version 599af129dbcfd4e85a2da9832c4ae59898e2f3a9) were removed.

Somatic variants were annotated using Ensembl Variant Effect Predictor (version 87)[52,53]. The tumour mutation burden was computed as the sum of all mutations per tumour divided by the total number of bases sequenced in the genome (45.54 Mb). Breast cancer driver mutations were defined as those genes identified in previous publications[54,55].

## Copy number calling

Genome binning and segmentation on low-pass sWGS data were performed using the R package QDNAseq (version 1.24)[56]. Binning was performed across 100-kb windows and counts corrected for GC-rich regions as well as poorly mappable regions. sWGS data from normal tissues were used to correct for technical and germline artefacts. Segmentation was performed using the circular binary segmentation algorithm implemented in the R package DNAcopy (version 1.60)[57]. Parental copy number quantification and estimation of tumour purity and ploidy were obtained using ASCAT (version 2.5.1)[58] using log ratios derived from QDNAseq and germline single-nucleotide polymorphisms obtained from HaplotypeCaller as input. As recommended by the authors, the technology parameter gamma was set to 1 for WES data.

## Clonal reconstruction

The CCF for each mutation was computed using the previously derived mathematical framework[16]:

$$CCF = \frac{VAF}{p} \times ((1-p)CN_{normal} + pCN_{tumour})$$

where VAF was the variant allele fraction for each mutation determined by exome sequencing, $p$ was the tumour purity (computed using ASCAT), $CN_{normal}$ was the germline copy number state and $CN_{tumour}$ was the total copy number state at the mutant locus in the tumour. Point estimates for CCF and confidence intervals were computed using a binomial distribution modelled by the binconf function from the Hmisc R package (version 4.4) and a mutation was classified as clonal if the CCF 95% confidence interval overlapped 1, with all other mutations classified as subclonal.

## Mutational signatures decomposition

Signature decomposition from the bulk exome sequencing mutation data was performed using the DeconstructSigs R package (version 1.8)[59], which uses the Wellcome Trust Sanger Institute Mutational Signature Framework as a reference and determines the linear combination of 30 pre-defined signatures by using a multiple logistic regression model with constraints to reconstruct the mutational profile of each tumour. Mutational signatures were solely identified in tumours with more than 10 mutations. To determine signature associations with response, each signature was $\log_2$ normalized using the exposure of signature 1 (age) as a reference. Associations between these normalized exposures and response were determined using logistic regression models.

## HRD quantification

The scarHRD R package (version 0.1.1)[18] was used to determine the levels of HRD present in the WES data, using the ASCAT allele-specific copy number as input. This tool inferred three components of HRD: telomeric allelic imbalance, LOH and the number of large-scale transitions, which were then summarized into an overall HRD score.

## HLA typing, identification of HLA LOH and neoantigen calling

HLA typing was performed on the normal tissue sequencing data using the Polysolver tool (version 4)[60], which inferred the four-digit HLA type for each sample by using a Bayesian classifier to determine genotype. LOH over the HLA class I locus was determined by using the LOHHLA tool (downloaded from https://bitbucket.org/mcgranahanlab/lohhla/src/master/ commit: 9d58c99)[21], using as input ASCAT tumour purity and HLA genotyping data from PolySolver (version 4). Statistically significant HLA alleles with a copy number of less than 0.5 were assumed to be undergoing LOH. Neoantigen calling was performed by using the pVAC-tools (version 1.5.4) cancer immunotherapy suite[61]. Mutations identified on exome sequencing were translated into corresponding mutant proteins and a list of potential neoantigenic fragments containing the mutant protein generated by using a sliding window approach across the mutated locus, retaining epitopes of lengths 8–11 amino acids. These potentially antigenic fragments were analysed for binding affinity to the HLA class I molecules using the prediction software NetMHCPan version 3[62], NetMHC version 4[63] and PickPocket version 1.1[64] bundled within the Immune Epitope Database resource[65]. Neoantigens with a binding affinity score of less than 500 nM and that had a higher binding affinity than the corresponding wild-type sequences were retained. Further downstream filtering was done by retaining neoepitopes generated by transcripts that had an expression greater than 1 TPM.

## iC10 classification

Classification of all tumours into one of the ten iC10 clusters[19,66] was performed using the iC10 R package (version 1.5)[20], which took cellularity-corrected copy number log ratios obtained from QDNAseq and voom-normalized gene expression counts derived from the RNA sequencing (RNA-seq) data as input. The iC10 classification of tumours that did not have RNA-seq data was determined using the copy number data only. Associations with response were visualized using the mosaic function from the vcd R package (version 1.4-7).

## RNA-seq pre-processing

FASTQ files for each sample generated from multiple sequencing lanes were merged and aligned using STAR version 2.5.2b[67], using an index generated from the GRCh37 decoy assembly of the human genome and a transcriptomic Gene Transfer Format (GTF) guide obtained from Ensembl Release 87. STAR was run in two-pass mode for sensitive novel junction discovery, in which the first pass performed a default mapping, and the second pass used the splice junctions detected in the first pass to perform a further round of alignment enhancement. This STAR BAM file was used for differential expression and transcript counting. For variant calling, the BAM files generated by STAR were processed as per GATK best practices guidelines: PCR and optical duplicates were marked using Picard MarkDuplicates and following this, the GATK tool SplitNCigarReads was used to split reads having N CIGAR elements in separate sequence reads. Local realignment around insertions and deletions was performed using RealignerTargetCreator and IndelRealigner, using a calibration set derived from the 1000 Genomes project[68–70]. Base quality recalibration across all variant sites was then performed using BaseRecalibrator. The tumour samples were sequenced to a median of 87 million reads.

## RNA variant calling

Germline variants identified on exome sequencing were filtered by removing multi-allelic variants, indels, as well as mutations for which the minimum depth was less than 30× across all samples. The remaining germline variants were subsequently genotyped across all RNA samples and comparisons were done across homozygous germline variants only. The percentage median concordance across samples derived from a matched patient was 100%, whereas unrelated samples had a median concordance of 60%. Somatic variants detected on exome sequencing were genotyped in the RNA GATK BAM by using HaplotypeCaller in GENOTYPE_GIVEN_ALLELES mode. Mutations present in all samples for one patient were concatenated together, and a VCF was generated to guide HaplotypeCaller local reassembly and variant calling.

## Gene and transcript abundance estimation

Gene expression estimation was performed on the STAR aligned BAM file using HTSeq (version 0.6.1p1)[71] in read strand-aware union

overlap resolution mode, where a read would only be assigned to a gene if it only overlapped within an exonic region of one gene, rather than multiple genes. Gene counts across all samples were merged into one counts matrix using R, and a trimmed mean of M-value (TMM) normalization performed across all samples using the edgeR R package (version 3.32.1)[72] to correct for composition biases and make the transcript counts comparable across all samples[73,74]. The library normalized counts were then transformed into fragment per kilobase millions (FPKMs) and then scaled to a total of a million counts, changing the unit of measure to transcripts per million (TPM)[75].

## Differential expression

To identify sets of genes that were highly or lowly expressed given a set of experimental conditions (such as pCR versus residual disease) (Fig. 3a), differential expression was performed on the gene raw counts data obtained as described above using edgeR[72,73]. The output of each model was a list of differentially expressed genes. Following the generation of a ranked list of differentially expressed genes for any comparison of interest, gene set enrichment was performed using the camera statistical method in edgeR; in brief, this method performed a competitive gene set test accounting for inter-gene correlation and tested whether genes were highly ranked relative to other genes in terms of differential expression[76]. As input to this gene set enrichment analysis (GSEA) method, the annotated gene sets provided within the MSigDB version 6.1 were used[22,77] (Fig. 3b). In addition, further enrichment over the Reactome database[23] (Extended Data Fig. 5) was performed using the ReactomePA R package (version v1.34)[78].

## GSEA

GSVA and ssGSEA were performed using the GSVA R package (version 1.34)[79] on (1) the GGI gene set[24], (2) the core embryonic stem-cell-like module[25] and (3) the STAT1 immune signature[31]. The log-transformed TMM normalized TPM counts were used as input to the GSVA package. A high GSVA score (Fig. 3f, Extended Data Fig. 8a) was defined as any score above the mean value. We computed the paclitaxel response metagene[26], as the difference in expression of a mitotic metagene (geometric mean of *BUB1B*, *CDK1*, *AURKB* and *TTK* TPM expression) and a ceramide metagene (geometric mean of *UGCG* and *CERT1* expression).

## Immune microenvironment characterization

The cytolytic activity score[27] was computed as the geometric mean of *GZMA* and *PRF1* (as expressed in TPM, 0.01 offset). Immune cell enrichment was performed using (1) MCPcounter[29] using voom-normalized RNA-seq counts as input, (2) enrichment over 14 cell types using 60 genes[28], using the log-transformed geometric mean of the TPM expression of cell-specific genes as input, and (3) *z*-score scaling of cancer immunity parameters[30] to classify four different immune processes (MHC molecules, immunomodulators, effector cells and suppressor cells), by generating *z*-score-normalized TPM gene expression for an input list of 162 genes. Heatmaps used to visualize the data were generated using the pheatmap R package (version 1.0.12) and unsupervised column hierarchical clustering based on the Euclidean distance performed. We used the TIDE algorithm (http://tide.dfci. harvard.edu)[32] to derive T cell dysfunction and exclusion metrics. The input to TIDE was a log$_2$-transformed TPM matrix of counts, which was normalized by subtracting the average log$_2$ expression of all genes. The interplay between proliferation and immune activation across the four RCB classes (shown in Extended Data Fig. 7f) was validated by performing GGI and STAT1 enrichment using a combined microarray dataset from the ISPY-I[10] (GSE25066 and GSE32603) and NCT00455533 (ref. [11]) (GSE41998) trials, which were chosen for similar neoadjuvant therapy regimens, availability of core biopsy gene expression and RCB classification.

## Digital pathology analysis

Whole-slide H&E images (scanned at a magnification of ×20) were analysed using CellExtractor v1.0, an open-source platform developed for high-throughput analyses of histopathological images. The code was written in Python and used the OpenCV and OpenSlide library. Initially, full-face H&E scanned images were divided into several subregions. Each subregion was processed to remove the background using an adaptive threshold method. A distance matrix was calculated for individual foreground objects to de-blend overlapping objects during the watershed segmentation process. The latter produced binary images of cell masks from which cellular features such as centroids, shape descriptors, and pixel intensities were estimated. These features were used to train a two-level support vector machine-based classifier. During the first level, spurious detections such as artefacts, dirt and pen marks were separated from genuine detections. This was followed by a second level of classification to identify cancer cells, stromal cells and lymphocytes based on a training set of objects selected by a pathologist (W.C.) of approximately 1,000 objects for each category. Finally, on the basis of these classes, descriptive statistical parameters such as cellular fractions and densities were estimated. For each detected cell, density was obtained based on counting the number of nearest neighbours approach, that is, the density within the distance to the *N*th nearest neighbour calculated as follows: $\text{Sigma}_N$ (pixel$^{-2}$) = $N/(\pi \times d_N^2)$ where $d_N$ was the distance to the *N*th nearest neighbour within a density-defining population. A value of $N = 50$ was used to estimate the density parameter[13]. To ensure that the estimated density was not biased towards our choice of density parameter ($N = 50$), we calculated the density for *N* in range of 40–60, with 5-step increments. The results remained the same and were therefore independent of the choice of the number of neighbours.

## Validation dataset

An external dataset comprising 75 patients treated with neoadjuvant therapy recruited to the Personalised Breast Cancer Programme (PBCP; research ethics reference: 18/EE/0251) study and the control arm of the ARTemis trial (research ethics reference: 08/H1102/104, EudraCT number: 2008-002322-11) was collated. All patients provided informed consent for sample collection and all participants consented to the publication of research results. These cases were selected due to the availability of DNA, RNA and digital pathology data. Clinical and molecular details for these 75 cases are summarized in Supplementary Table 5.

## Statistical testing

All statistical tests in the exploratory analysis were performed using R version 4.0.3 and associated packages. All statistical tests described in this work were two-sided. Unless otherwise specified, all statistical comparisons were performed using cases that attained pCR as a comparator. Tests involving comparisons of distributions were done using 'wilcox.test' unless otherwise specified. Ordinal logistic regression models used the ordered RCB variable (pCR > RCB-I > RCB-II > RCB-III) as a response variable to determine monotonic associations and were modelled using the polr function from the MASS R package (version 7.3-54). To determine features associated with response, only cases that received at least one cycle of neoadjuvant chemotherapy and one cycle of anti-HER2 therapy (if HER2$^+$) were used in the comparisons to avoid the confounding effect of suboptimal exposure to neoadjuvant therapy on response.

## Derivation of a predictive model for relapse

**Dataset and model training.** The TransNEO dataset was used to train the machine learning pCR classification models. Hyperparameters were optimized using fivefold cross-validation in the training set to maximize the area under the receiver operating characteristic (AUC ROC) curve. The rest of the parameters were determined by setting

the hyperparameters to their optimal values and refitting to the entire training cohort. To ensure robustness, we repeated the optimization process five times with different cross-validation seeds, effectively training five alternative predictors. Together, these five predictors constituted what we call the 'model': model predictions for new data are obtained by averaging the scores produced by the five predictors. Once trained and frozen, models were independently validated on an external dataset composed of $n = 75$ patients from the PBCP and ARTemis cohorts described previously.

**Predictor architecture.** The machine learning framework was built on Python (version 3.7.4) using the following libraries: scikit-learn (version 0.21.2), numpy (version 1.16.4), scipy (version 1.3), pandas (version 0.24.2) within a Singularity container (version 2.4.6-dist). Each predictor was built as an ensemble of three scikit-learn pipelines; in other words, the response prediction was calculated as the average of the scores produced by the three classification pipelines. Each pipeline contained four steps: collinearity removal, $k$-best feature selection, scaling and classification. The first step removed all features with a mutual Pearson correlation above 0.8, retaining only the one with the highest correlation with the response variable. The second step removed all features that were not ranked within the top $k$ according to their ANOVA $F$-value with respect to the binary response variable. The third step applied $z$-score scaling to the remaining features. The fourth step was the classification step, which consisted of a logistic regression[80] in the first pipeline, a support vector classifier[81] in the second pipeline, and a random forest[82] in the third pipeline. All hyperparameters were optimized using a randomized 1,000-step fivefold cross-validation search to maximize the AUC ROC curve. Logistic regression was implemented with elastic net regularization and SAGA solver, with C parameters between $10^{-3}$ and $10^{3}$, and L1 ratios between 0.1 and 1. The support vector classifier was allowed to have either radial basis function, sigmoid or linear kernels, with gamma parameters between $10^{-9}$ and $10^{-2}$, and C parameters between $10^{-3}$ and $10^{3}$. Finally, the random forests were allowed to have between 5 and 100 (or the maximum number of) estimators, maximum features between 5% and 70% of the total, and minimum samples per split between 2 and 15. The final values of the hyperparameters obtained through the optimization procedure can be found in the Supplementary Material.

**Feature definitions.** Models were trained on a combination of clinical, DNA, RNA, digital pathology and treatment features, as shown in Fig. 4a. Differences in treatment were captured using one-hot-encoded variables assessing whether the patient did or did not receive anthracycline or anti-HER2 treatment. A further set of variables captured whether taxane or anthracycline were given first. The complete list of features and their Spearman correlation matrix can be found in Supplementary Table 4 and Extended Data Fig. 9a, respectively. The order in which features were added in successive models was determined by how widely available they typically are. Although the information required for treatment variables is normally accessible, they are highly correlated with HER2 status, and are therefore included mainly as a cautionary control mechanism. For the sake of the simplicity of the models, they were the last features to be added.

**Data cleaning.** In the training set, one patient who had clinically unevaluable tumour size was assumed to have a volume 10% larger than the largest present in the cohort. Four patients who were HER2$^+$ who only received one cycle of trastuzumab, and two patients who were HER2$^-$ who had only received one chemotherapy cycle were removed from the training set. In the external validation datasets, missing treatment features were set to zero.

**Testing.** Models were applied on the test cohort and their respective ROC curves and AUCs were evaluated. In Fig. 4d, the standard deviation of the AUCs obtained in the training cross-validation (included as an optimistic performance estimation) was compared to the nominal test AUCs and the standard deviation of the AUCs obtained from 100 bootstrap replicas of the test datasets. In addition, 95% confidence intervals on each test AUC were obtained using the DeLong test[83] (Extended Data Fig. 9e). Adding digital pathology introduced a slight degradation of the performance due to the significant difference in the lymphocytic density observed in the training versus the external validation cohorts (Extended Data Fig. 9f). Precision-recall curves, average precision scores and areas under the precision-recall curve were obtained using standard sklearn implementations (Extended Data Fig. 9g).

**Feature importance.** Feature importances were determined for each algorithm (random forest, support vector classifier and logistic regression) after refitting on the full training cohorts. For consistency, we used an algorithm-agnostic methodology based on dropping each of the input features. We quantified the resulting change in AUC by means of a $z$-score, $z^i = \frac{|AUC_{nominal} - AUC^i_{drop}|}{\sigma(AUC_{nominal} - AUC_{drop})}$, where $z^i$ represents the $z$-score significance of the $i$th feature, and $\sigma$ is the standard deviation of all the AUC changes. In Fig. 4b, we show the average $z$-score significances averaged across the three algorithms. In Extended Data Fig. 9b, we calculate signed $z$-score significances by removing the absolute value from the definition. The sign indicates whether the feature was adding value to the prediction (negative sign) or harming it (positive sign). In addition, the full list of features selected after the collinearity reduction and univariable selection steps for all the different models, as well as the logistic regression coefficients, can be found in the Supplementary Material.

### Reporting summary

Further information on research design is available in the Nature Research Reporting Summary linked to this paper.

## Data availability

DNA and RNA sequencing data have been deposited at the European Genome-Phenome Archive (EGA), which is hosted by the EBI and the CRG, under accession number EGAS00001004582.

## Code availability

The R and Python source code used to run the analyses described in the article and to generate all figures is available at: https://github.com/cclab-brca/neoadjuvant-therapy-response-predictor.

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

**Acknowledgements** S.-J.S. was supported by a Wellcome Trust PhD Clinical Training Fellowship (grant number 106566/Z/14/Z). M.C.-O. was supported by a Junior Research Fellowship from Trinity College, Cambridge, and a Borysiewicz Fellowship from the University of Cambridge. F.M. was supported by funding from Cancer Research UK (CRUK) (grant numbers A17197 and A19274). O.M.R. was supported by the NIHR Cambridge Biomedical Research Centre (BRC-1215-20014) and the Medical Research Council (UK; MC_UU_00002/16). C.C. was supported by funding from CRUK (grant numbers A17197, A27657 and A29580), an NIHR Senior Investigator Award (grant number NF-SI-0515-10090) and a European Research Council Advanced Award (grant number 694620). The ARTemis molecular profiling dataset was funded in part by an unrestricted academic grant from F. Hoffman La Roche (grant administered by the University of Cambridge). The Programme received infrastructure funding from the CRUK Cambridge Centre and the Mark Foundation Institute for Integrated Cancer Medicine. We are grateful for the generosity of all the patients that donated samples for analysis; all the staff at the Cambridge Breast Cancer Research Unit for facilitating the collection and processing of samples; and the CRUK Cambridge Institute Core Facilities (Genomics, Bioinformatics, Histopathology and Biorepository) for support during the execution of this project.

**Author contributions** S.-J.S. and C.C. conceived the study, led data analysis and wrote the manuscript. Tumour processing was led by S.-J.S. with input from S.-F.C., H.A.B. and W.M. E.P. and J.T. provided histopathology expertise and calculated the RCB index. S.-J.S. created the bioinformatics analysis pipeline, performed all DNA and RNA analyses and identified univariable associations with response. W.C. and A.D. generated the digital pathology lymphocytic infiltration estimates. M.C.-O. and S.-J.S. developed and validated the machine learning models. O.M.R., P.D.P. and F.M. provided statistical advice and expertise. S.-J.D. wrote the TransNEO protocol. C.C. and J.E.A. contributed data from the Personalised Breast Cancer Programme for validation. H.M.E., J.E.A., J.D., L.Hiller, J.T., D.A.C., J.M.S.B., C.C. and L.Hayward are members of the ARTemis trial management group and contributed the data from the control arm of the trial for validation. All authors read and approved the manuscript.

**Competing interests** C.C. is a member of the iMED External Science Panel for AstraZeneca, a member of the Scientific Advisory Board for Illumina and a recipient of research grants (administered by the University of Cambridge) from Genentech, Roche, AstraZeneca and Servier. M.C.-O. has received research funding from Lilly. All other authors declare no competing interests.

**Additional information**
**Correspondence and requests for materials** should be addressed to Carlos Caldas.

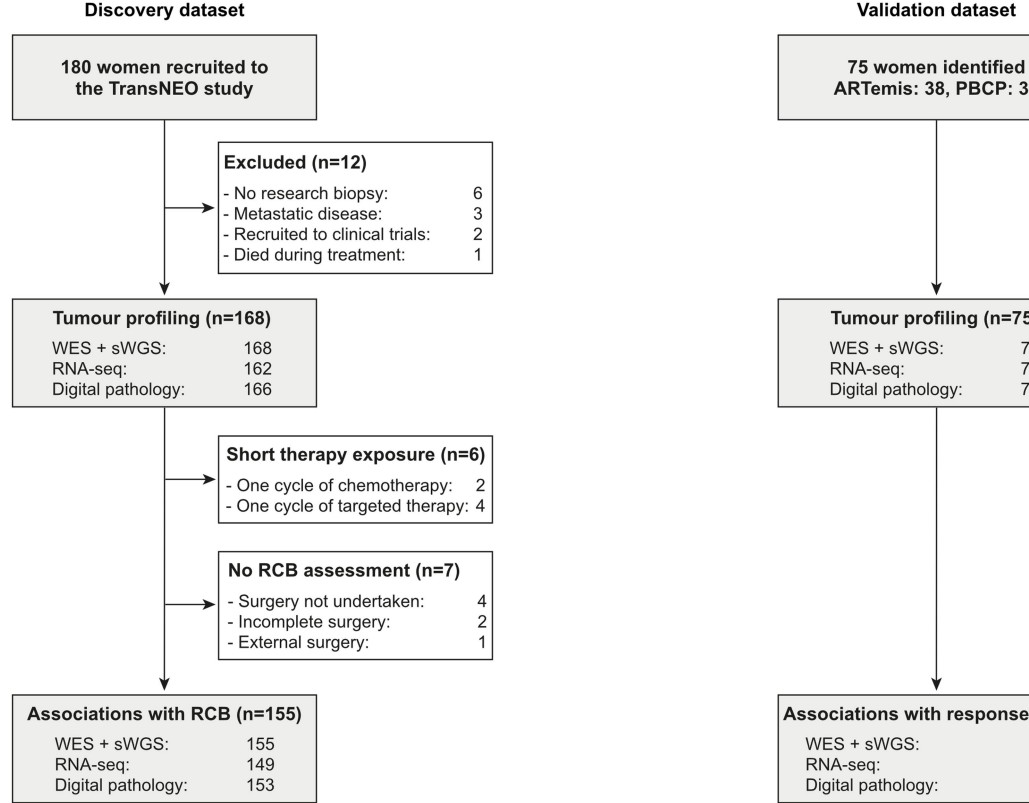

**Extended Data Fig. 1 | Summary of cases analysed within this study.** 180 women were recruited to the TransNEO neoadjuvant breast cancer study. Tumour profiling was performed in 168 cases and associations with response identified in 155 cases who received more than one cycle of neoadjuvant chemotherapy or targeted therapy. 147 cases had a complete molecular/digital pathology dataset, received more than one cycle of chemotherapy and targeted therapy and had an RCB assessment available: data from these cases were used to build a machine learning predictor of response to neoadjuvant therapy. Validation was performed across a cohort of 75 cases recruited to the ARTemis and Personalised Breast Cancer (PBCP) studies.

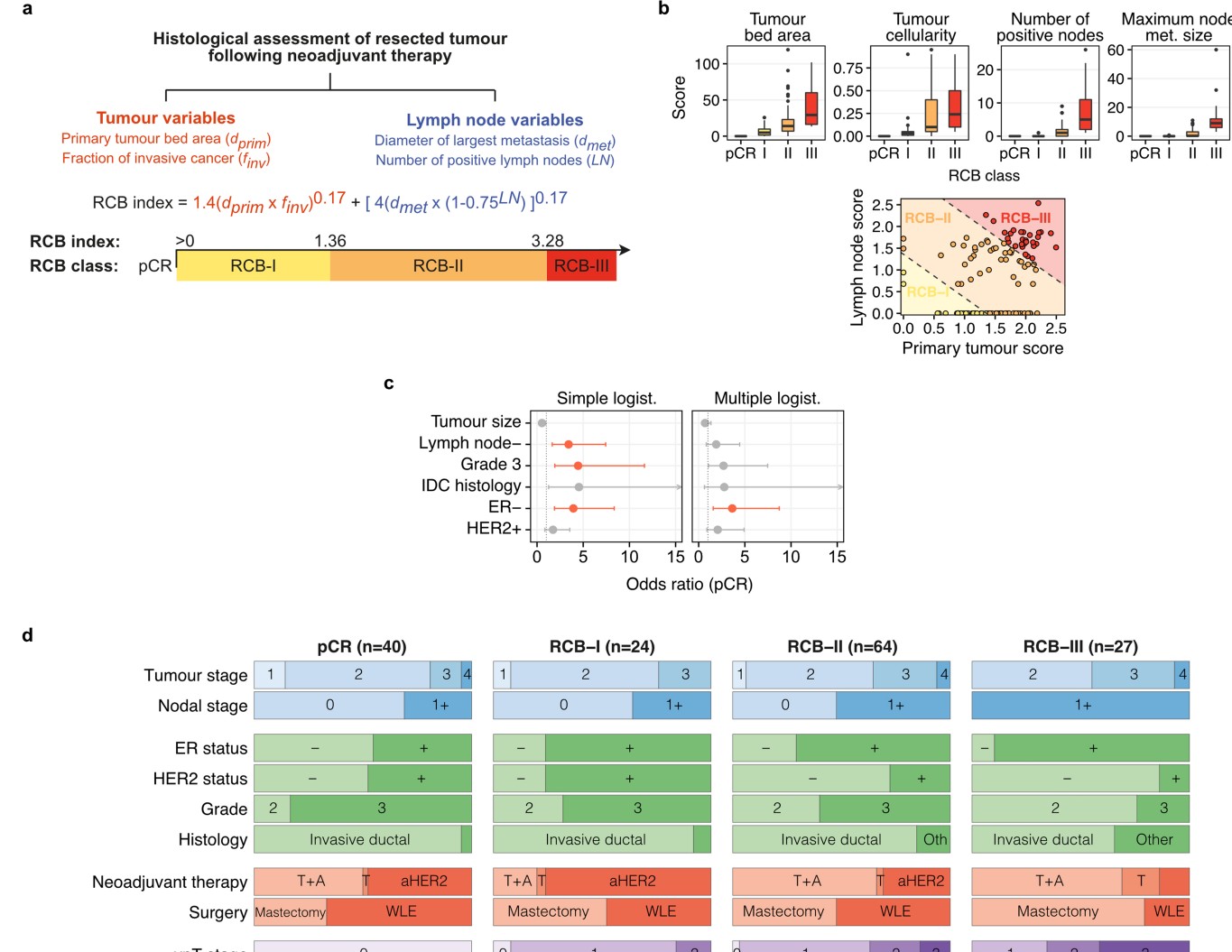

**Extended Data Fig. 2 | Calculation of the Residual Cancer Burden index and associations between clinical features and response. a**, Tumour and lymph node histological features used to calculate the continuous Residual Cancer Burden (RCB) index and categorical RCB class. Increasing RCB index denotes increasing burden of residual disease post-neoadjuvant therapy and increasing chemoresistance. **b**, Top: Box plots showing distribution of tumour and lymph node histological features in $n = 161$ cases with clinical data and RCB assessment across the RCB classes. The box bounds the interquartile range divided by the median, with the whiskers extending to a maximum of 1.5 times the interquartile range beyond the box. Outliers are shown as dots. Bottom: distribution of primary tumour score and lymph node score across RCB classes. **c**, Associations of clinical variables with pCR using simple and multiple logistic regression. Significant associations ($P < 0.05$, logistic regression) are shown in red. The measure of centre is the parameter estimate and error bars represent 95% confidence intervals. **d**, Distribution of tumour features across RCB classes: pre-operative staging (blue), pre-operative histological features (green), neoadjuvant therapy (red, T: taxane, A: anthracycline, aHER2: anti-HER2 therapy), surgical approach (red, WLE: wide local excision), post-operative tumour (ypT) and nodal (ypN) staging and lymphovascular invasion (purple) and PAM50 subtypes (yellow, A: Luminal A, B: Luminal B, Ba: Basal, H: HER2-enriched, N: Normal-like, U: Unknown). Tumours with RCB assessment and adequate therapy exposure only included (more than 1 cycle of chemotherapy or anti-HER2 therapy received, $n = 155$).

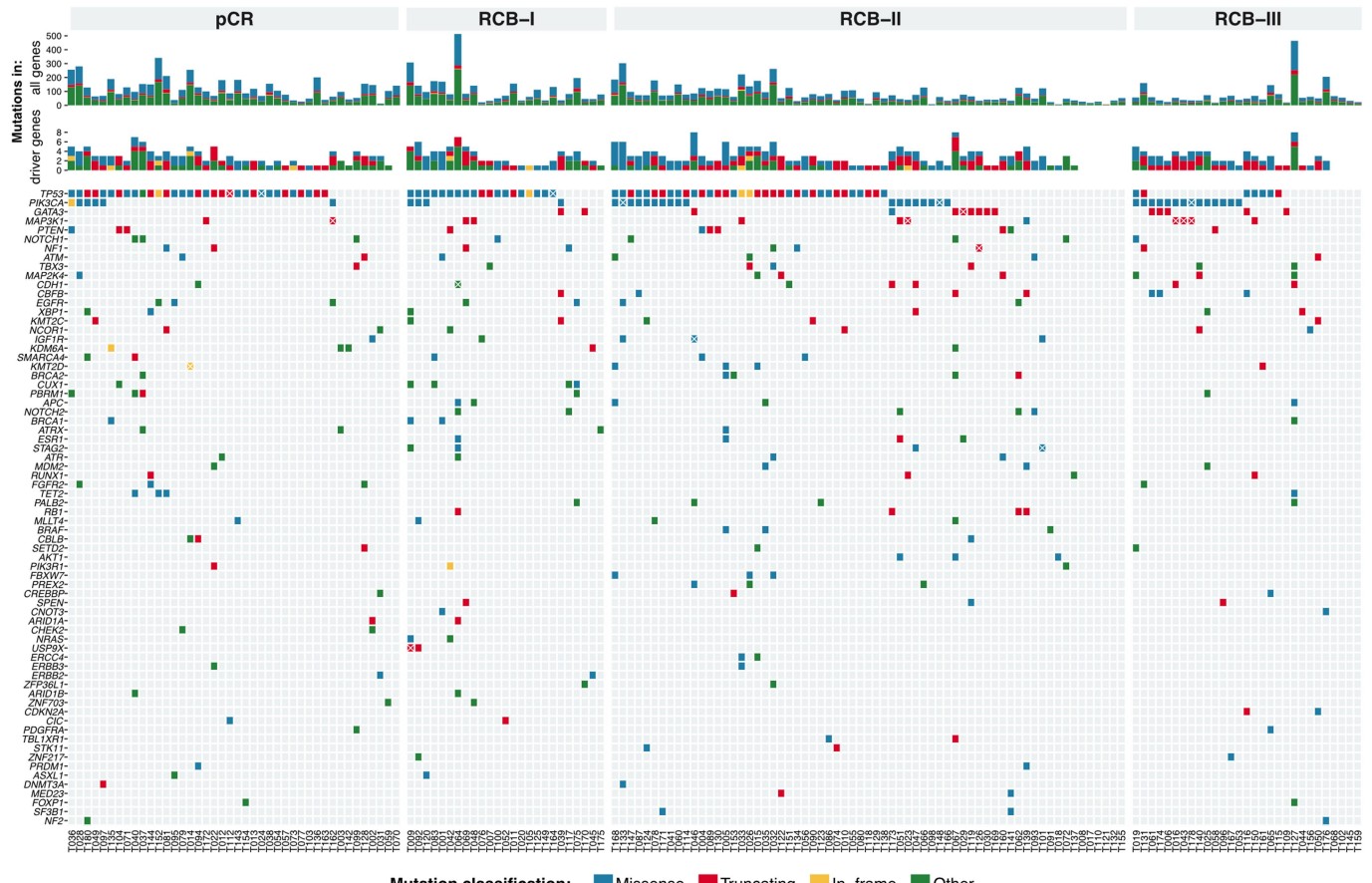

**Extended Data Fig. 3 | The somatic mutational driver landscape of tumours prior to neoadjuvant therapy.** Oncoprint showing somatic mutations in breast cancer driver gene identified using WES. Cases classified by RCB class. Multiple mutations in a case are denoted by a white ×. Truncating mutations (red) include nonsense, splice site and frame shift insertions and deletions. In-frame mutations (yellow) include in-frame insertions and deletions. Other mutations (green) include silent exonic mutations, 3' and 5' UTR flank mutations and intronic mutations.

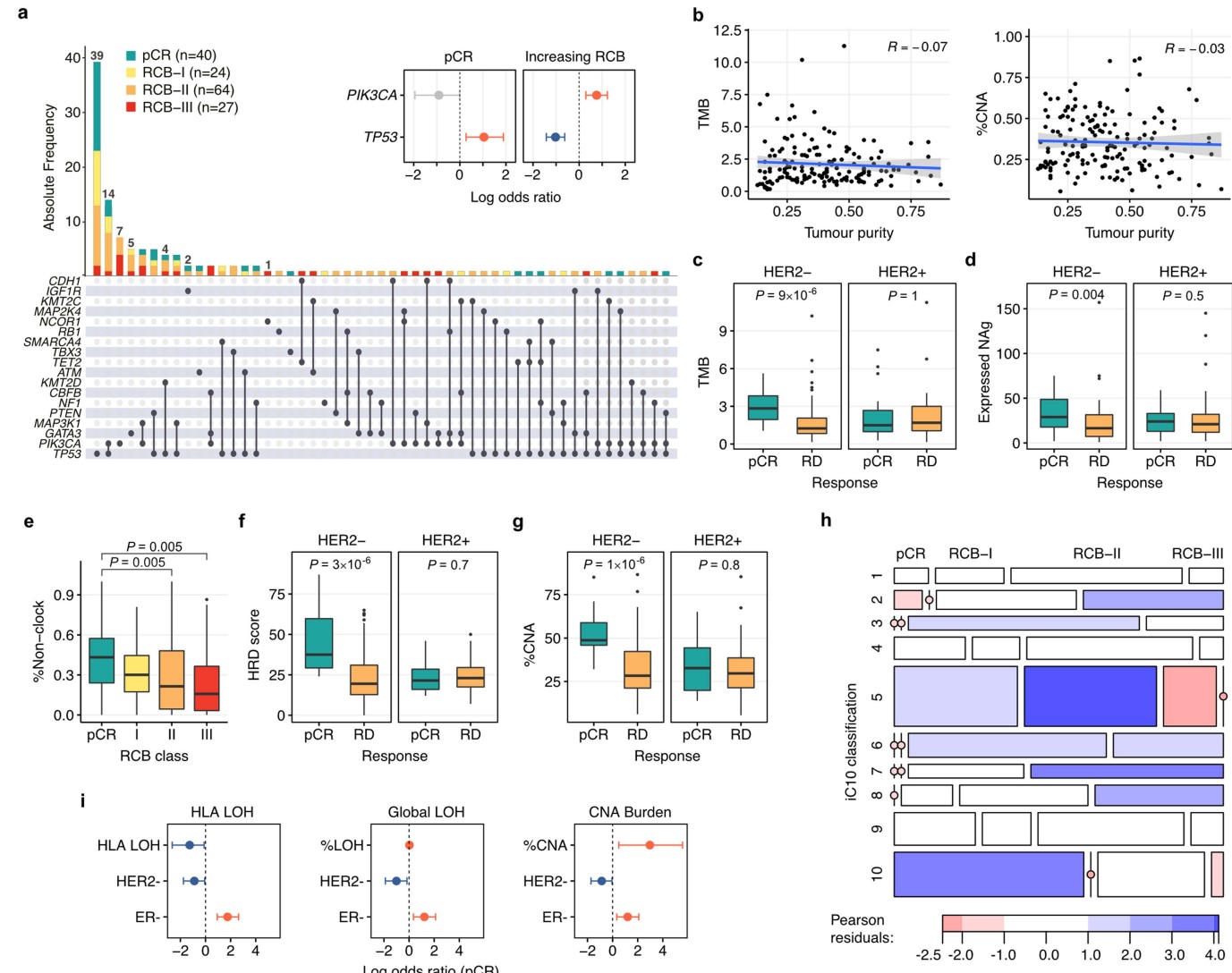

**Extended Data Fig. 4 | Further associations between genomic features and response to neoadjuvant therapy. a**, Interaction plot showing co-occurrence of non-silent driver gene mutations and response. Associations between *TP53* and *PIK3CA* mutations and response shown in inset (logistic regression, red: positive, blue: negative, grey: not significant, error bars represent 95% confidence intervals). **b**, Pearson's product-moment correlations (R) between tumour purity and (left) tumour mutation burden and (right) %CNAs. The shaded area, in grey, represents the 95% confidence interval. **c**, Box plots showing associations between TMB and response, stratified by HER2 status. **d**, Box plots showing association between expressed neoantigen (NAg) load and response, stratified by HER2 status. **e**, Box plot showing monotonic association (*P* = 0.005, ordinal logistic regression) between exposure of non-clock signatures and RCB class. **f**, Box plots showing associations between HRD score and response, stratified by HER2 status. **g**, Box plots showing associations between %CNA and response, stratified by HER2 status. **c**–**g**, The box bounds the interquartile range divided by the median, with the whiskers extending to a maximum of 1.5 times the interquartile range beyond the box. Outliers are shown as dots. Wilcoxon rank sum tests, all P values two-sided. Number of cases analysed (*n*) = 155 (HER2- pCR = 22, RD (residual disease) = 76; HER2+ pCR = 18, RD = 39)). **h**, Associations between RCB class and iC10: Pearson residuals indicate overrepresentation of iC10 subtype with response (blue: overrepresentation, red: underrepresentation). **i**, Associations between HLA LOH, global LOH and global copy number alterations with pCR (logistic regression, red: positive association, blue: negative association). The measure of centre is the parameter estimate and error bars represent 95% confidence intervals.

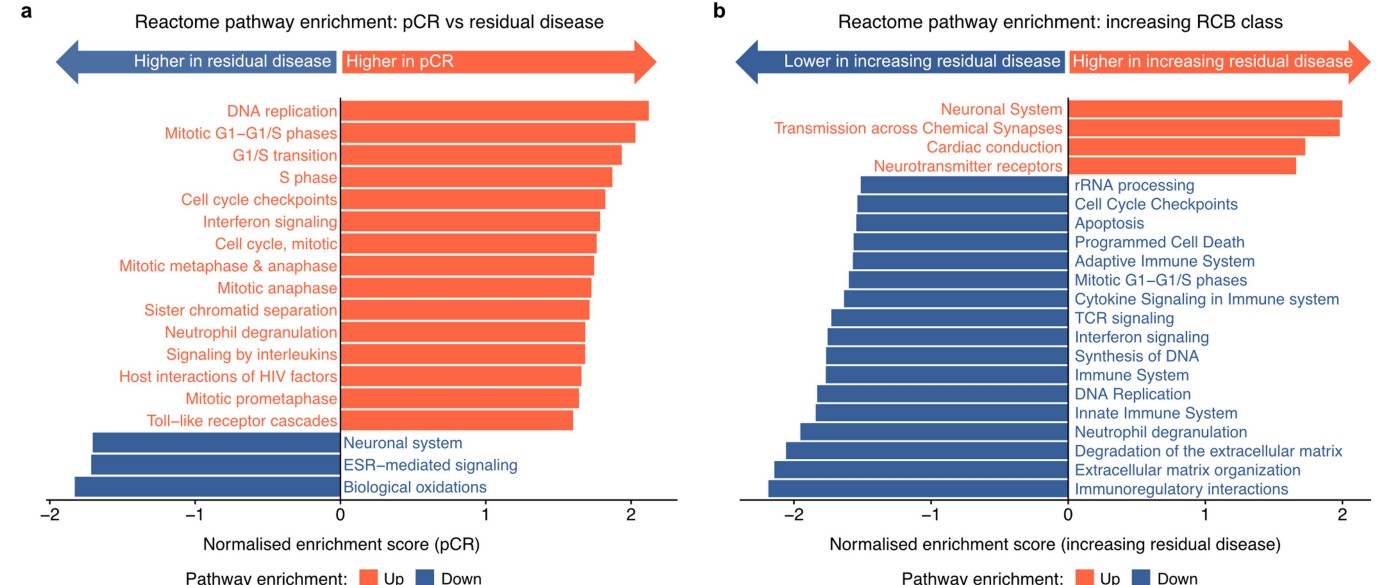

**Extended Data Fig. 5 | Reactome pathways associated with response to neoadjuvant therapy. a**, **b**, Reactome pathway enrichment showing pathways associated with (**a**) pCR versus residual disease, (**b**) degree of residual disease following neoadjuvant therapy.

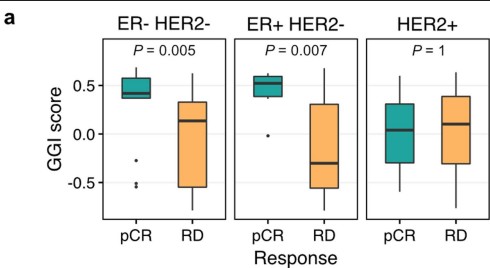

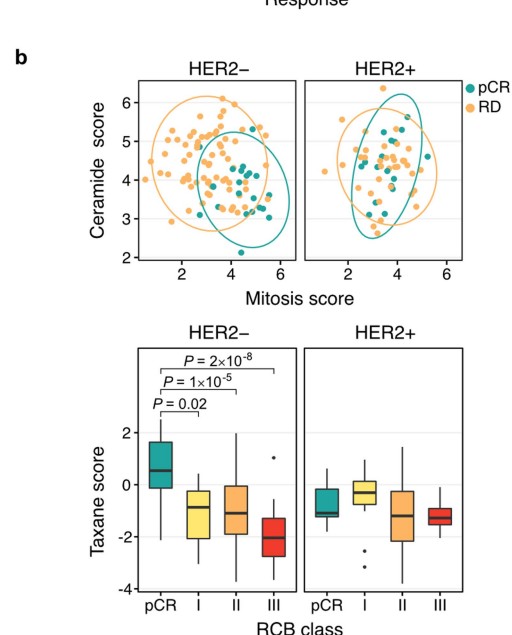

**Extended Data Fig. 6 | Associations between tumour proliferation and response. a**, Box plots showing associations between proliferation (GGI) GSVA scores across ER/HER subtypes. **b**, Top: Scatter plots showing the distribution of the mitotic and ceramide score components of a taxane response metagene within the HER2- and HER2+ cohorts. Bottom: Box plots showing association of the combined taxane response metagene score within the HER2- and HER2+ cohorts. In **a**, **b**, the box bounds the interquartile range divided by the median, with the whiskers extending to a maximum of 1.5 times the interquartile range beyond the box. Outliers are shown as dots. Two-tailed Wilcoxon rank sum tests. Number of cases (*n*): ER-HER2-: 37, ER+HER2-: 57, HER2+: 55.

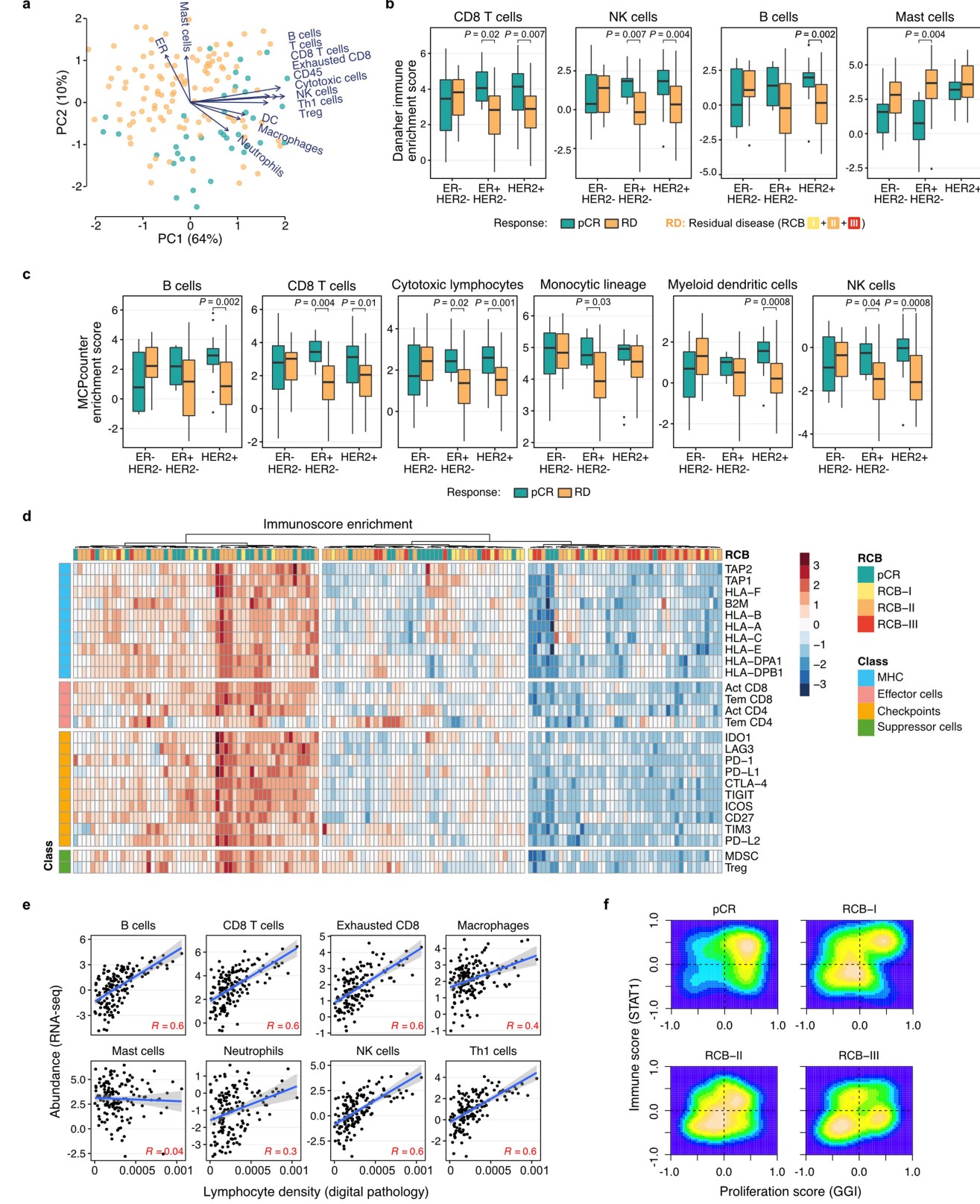

**Extended Data Fig. 7** | See next page for caption.

**Extended Data Fig. 7 | The relationship between tumour immune microenvironment and response. a**, PCA analysis on the abundance of tumour immune microenvironment components obtained through the deconvolution of RNA-seq data using Danaher's immune signatures (number of cases ($n$): pCR (green) = 39, RD (orange) = 110). **b, c**, Box plots showing associations between response and (**b**) Danaher immune cell enrichment and (**c**) MCPcounter immune cell enrichment across ER/HER subtypes. The box bounds the interquartile range divided by the median, with the whiskers extending to a maximum of 1.5 times the interquartile range beyond the box. Outliers are shown as dots. Two-tailed Wilcoxon rank sum tests. Number of cases ($n$): ER-HER2-: 37, ER+HER2-: 57, HER2+: 55. **d**, Heatmap showing unsupervised clustering of cancer immunity parameters across $n = 149$ cases with RNA sequencing data. **e**, Scatter plot showing association between computationally derived lymphocyte density and immune cell enrichment using Danaher's immune signatures across $n = 147$ cases with digital pathology and RNA sequencing data. Pearson's product-moment correlations (R) shown. The shaded area, in grey, represents the 95% confidence interval. **f**, 2D density plot validating relationship between GGI and STAT1 GSVA across RCB subgroups in two external microarray gene sets comprising 457 cases.

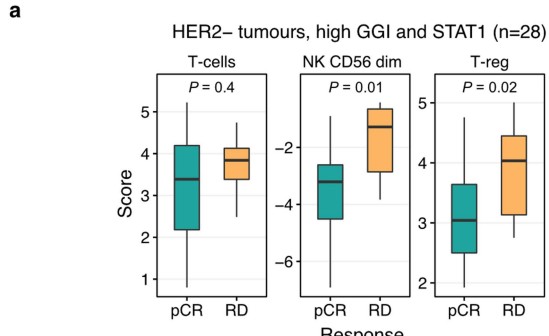

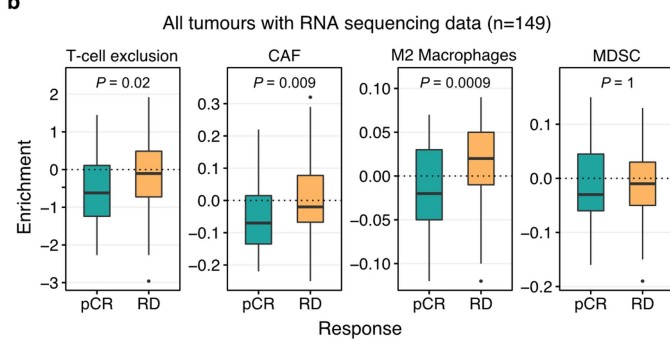

**Extended Data Fig. 8 | T-cell dysfunction and exclusion. a,** Box plots showing enriched inhibitory immune cell types (Danaher gene sets) in HER2- tumours with high GGI and STAT1 (number of cases (*n*): pCR = 12, RD = 16). **b,** Box plots showing association between components of T-cell exclusion score and response (number of cases (*n*): pCR = 39, RD = 110). CAF: Cancer associated fibroblasts, MDSC: Myeloid-derived suppressor cells. In **a**, **b**, the box bounds the interquartile range divided by the median, with the whiskers extending to a maximum of 1.5 times the interquartile range beyond the box. Outliers are shown as dots. Two-tailed Wilcoxon rank sum tests.

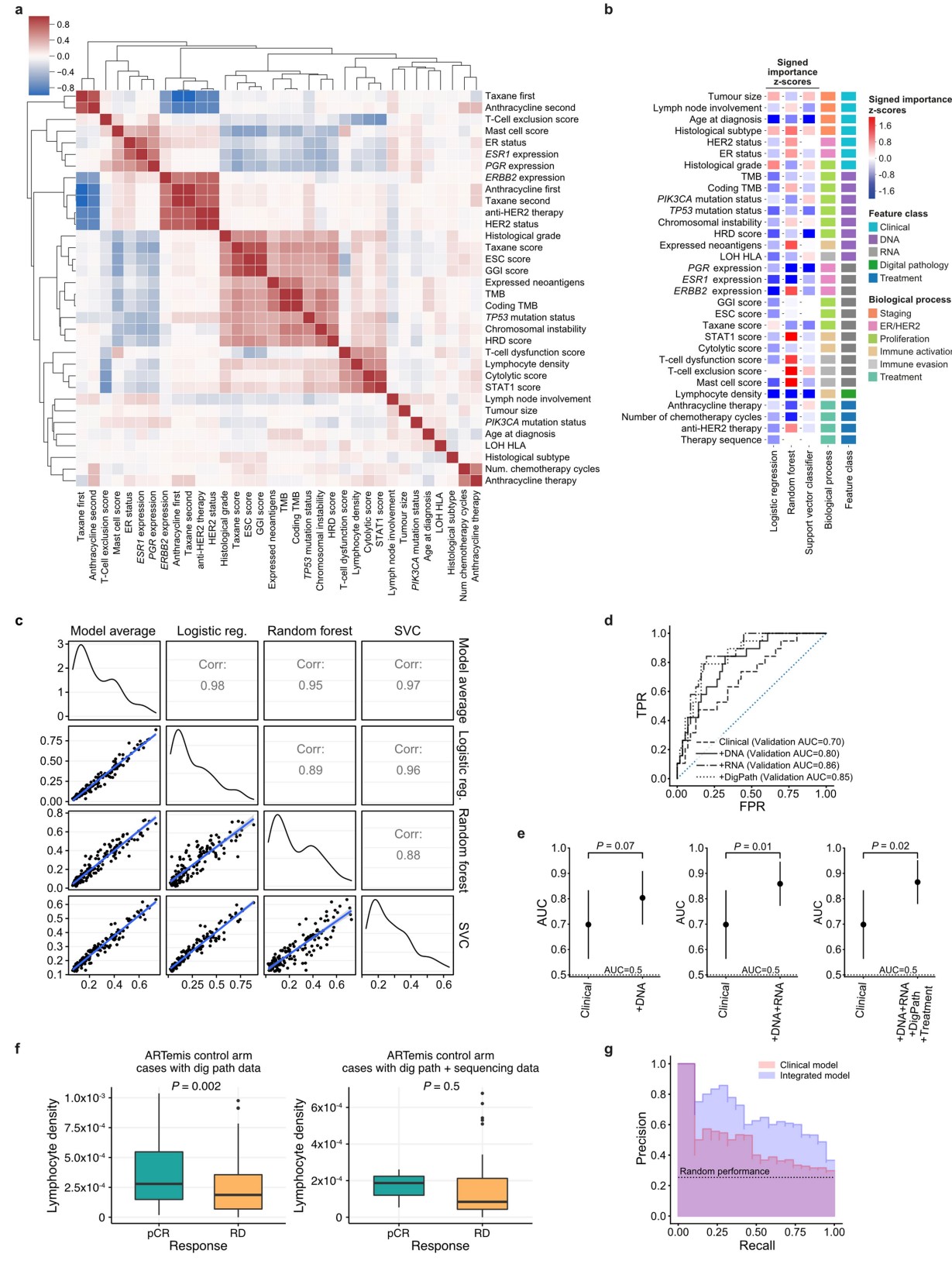

**Extended Data Fig. 9** | See next page for caption.

**Extended Data Fig. 9 | Machine learning model performance. a**, Correlation plot showing the results of unsupervised clustering between all the features explored. **b**, Signed feature importance split by algorithm. Negative numbers (blue) signify a decrease in AUC as a result of dropping, and therefore indicate that the feature improves the performance. **c**, Correlation of the three classification pipeline scores across the training dataset. Two-sided $P$ values of all correlations $< 2.2 \times 10^{-16}$. **d**, Receiver-operating characteristic curves for the clinical and integrated models applied on the external validation cohort. **e**, Comparison between AUCs of the clinical model and models with different levels of data integration. The measure of centre is the parameter estimate and error bars represent 95% DeLong confidence intervals. **f**, Association between lymphocyte density and treatment response in ARTemis patients with digital pathology and sequencing data (right, $n = 38$ cases) vs. patients with only digital pathology available (left, $n = 313$ cases). The box bounds the interquartile range divided by the median, with the whiskers extending to a maximum of 1.5 times the interquartile range beyond the box. Outliers are shown as dots. P values obtained from Wilcoxon rank sum tests. **g**, Precision-recall curves of the clinical and fully integrated models applied on the test cohorts. The average precision values are 0.46 (clinical model) and 0.68 (fully integrated model). The areas under the precision-recall curves are 0.43 (clinical model) and 0.67 (fully integrated model).

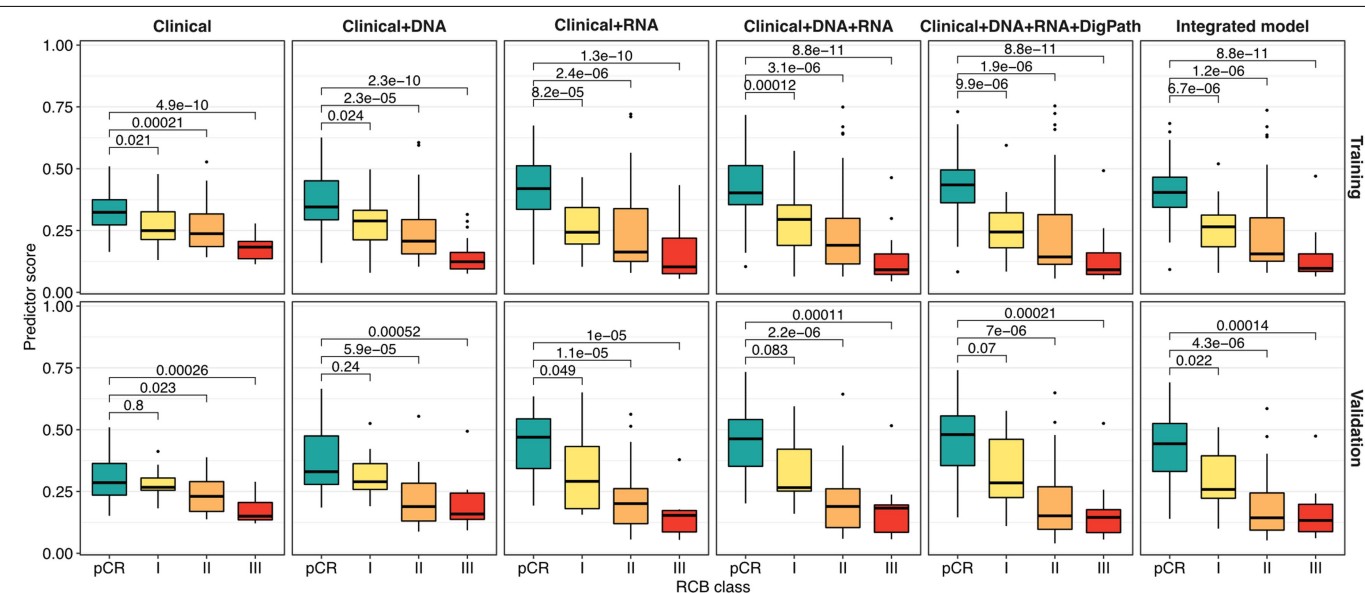

**Extended Data Fig. 10 | Predictor score ordinally associated with RCB class.** Box plots showing the distribution of predictor scores obtained by the six models across RCB classes in both training (*n* = 147 cases) and validation (*n* = 75 cases) sets. The box bounds the interquartile range divided by the median, with the whiskers extending to a maximum of 1.5 times the interquartile range beyond the box. Outliers are shown as dots. *P* values two-sided and obtained from FDR-corrected Wilcoxon rank sum tests.

# Reporting Summary

## Statistics

For all statistical analyses, confirm that the following items are present in the figure legend, table legend, main text, or Methods section.

| n/a | Confirmed | |
|---|---|---|
| ☐ | ☒ | The exact sample size (*n*) for each experimental group/condition, given as a discrete number and unit of measurement |
| ☐ | ☒ | A statement on whether measurements were taken from distinct samples or whether the same sample was measured repeatedly |
| ☐ | ☒ | The statistical test(s) used AND whether they are one- or two-sided<br>*Only common tests should be described solely by name; describe more complex techniques in the Methods section.* |
| ☐ | ☒ | A description of all covariates tested |
| ☐ | ☒ | A description of any assumptions or corrections, such as tests of normality and adjustment for multiple comparisons |
| ☐ | ☒ | A full description of the statistical parameters including central tendency (e.g. means) or other basic estimates (e.g. regression coefficient) AND variation (e.g. standard deviation) or associated estimates of uncertainty (e.g. confidence intervals) |
| ☐ | ☒ | For null hypothesis testing, the test statistic (e.g. *F*, *t*, *r*) with confidence intervals, effect sizes, degrees of freedom and *P* value noted<br>*Give P values as exact values whenever suitable.* |
| ☒ | ☐ | For Bayesian analysis, information on the choice of priors and Markov chain Monte Carlo settings |
| ☒ | ☐ | For hierarchical and complex designs, identification of the appropriate level for tests and full reporting of outcomes |
| ☒ | ☐ | Estimates of effect sizes (e.g. Cohen's *d*, Pearson's *r*), indicating how they were calculated |

*Our web collection on statistics for biologists contains articles on many of the points above.*

## Software and code

Policy information about availability of computer code

| | |
|---|---|
| Data collection | Clinical data was collected in Microsoft Excel (as part of the office 365 suite) by data managers, and then converted into R objects using the R statistical framework (v 4.0.3) |
| Data analysis | List of software used:<br><br>ANNOVAR: version 599af129dbcfd4e85a2da9832c4ae59898e2f3a9<br>ASCAT: version 2.5.1<br>bcl2fastq2: version 2.17<br>CellExtractor: version v1.0<br>Ensembl Variant Effect Predictor: version 87<br>FastQC: version 0.11.7<br>Genome Analysis Toolkit (GATK): version 4.1.4. Tools used: BaseRecalibrator,  CreateSomaticPanelOfNormals, FilterMutectCalls, HaplotypeCaller, IndelRealigner, Mutect2, RealignerTargetCreator, SplitNCigarReads, VariantRecalibrator<br>HTSeq: version 0.6.1p1<br>LOHHLA: https://bitbucket.org/mcgranahanlab/lohhla/src/master/ commit 9d58c99<br>Microsoft Excel: office 365 version<br>Novoalign and Novosort: version 3.2.13<br>NetMHC: version 4<br>NetMHCPan: version 3<br>Picard: version 2.17.0. Tools used: CalculateHSMetrics, MarkDuplicates<br>PickPocket: version 1.1<br>Polysolver: version 4 |

pVAC-tools: version 1.5.4
Singularity: version 2.4.6-dist
STAR: version 2.5.2b
TIDE: http://tide.dfci.harvard.edu

R version 4.0.3 and associated packages:
• DeconstructSigs: version 1.8
• DNAcopy: version 1.60
• edgeR: version 3.32.1
• GSVA: version 1.34
• Hmisc version 4.4
• iC10: version 1.5
• MASS: version 7.3-54
• MCPcounter: version 1.2.0
• pheatmap: version 1.0.12
• QDNAseq: version 1.24
• ReactomePA: version 1.34
• scarHRD: version 0.1.1
• vcd: version 1.4-7

Python version 3.7.4 and associated packages:
• Numpy: version 1.16.4
• Scipy: version 1.3
• Scikit-learn: version 0.21.2
• Pandas: version 0.24.2

For manuscripts utilizing custom algorithms or software that are central to the research but not yet described in published literature, software must be made available to editors and reviewers. We strongly encourage code deposition in a community repository (e.g. GitHub). See the Nature Portfolio guidelines for submitting code & software for further information.

# Data

Policy information about availability of data

All manuscripts must include a data availability statement. This statement should provide the following information, where applicable:

- Accession codes, unique identifiers, or web links for publicly available datasets
- A description of any restrictions on data availability
- For clinical datasets or third party data, please ensure that the statement adheres to our policy

DNA and RNA sequence data have been deposited at the European Genome-phenome Archive (EGA), which is hosted by the EBI and the CRG, under accession number EGAS00001004582 (https://ega-archive.org).

Individual raw data sets are available in Supplementary Tables 1–4.

The R and Python source code used to run the analyses described in the manuscript and to generate all figures is available at: https://github.com/cclab-brca/neoadjuvant-therapy-response-predictor

The following gene sets are referenced within the manuscript:

1. Molecular Signatures Database (MSigDB) Hallmarks gene set (version 6.1). Downloaded from: https://www.gsea-msigdb.org/gsea/msigdb/
2. Genomic Grade Index (GGI) gene set. Reference: Sotiriou, C. et al. Gene expression profiling in breast cancer: understanding the molecular basis of histologic grade to improve prognosis. J. Natl. Cancer Inst. 98, 262–72 (2006).
3. Core Embryonic stem cell (ESC)-like module. Reference: Wong, D. J. et al. Module map of stem cell genes guides creation of epithelial cancer stem cells. Cell Stem Cell 2, 333–44 (2008).
4. STAT1 immune signature. Reference: Desmedt, C. et al. Biological processes associated with breast cancer clinical outcome depend on the molecular subtypes. Clin. Cancer Res. 14, 5158–65 (2008).
5. Paclitaxel response metagene. Reference: Juul, N. et al. Assessment of an RNA interference screen-derived mitotic and ceramide pathway metagene as a predictor of response to neoadjuvant paclitaxel for primary triple-negative breast cancer: a retrospective analysis of five clinical trials. Lancet. Oncol. 11, 358–65 (2010).
6. Cytolytic activity (CYT) score. Reference: Rooney, M. S., Shukla, S. A., Wu, C. J., Getz, G. & Hacohen, N. Molecular and genetic properties of tumors associated with local immune cytolytic activity. Cell 160, 48–61 (2015).
7. Danaher immune gene sets. Reference: Danaher, P. et al. Gene expression markers of Tumor Infiltrating Leukocytes. J. Immunother. Cancer 5, 18 (2017).
8. Immunoscore gene sets. Reference: Charoentong, P. et al. Pan-cancer Immunogenomic Analyses Reveal Genotype-Immunophenotype Relationships and Predictors of Response to Checkpoint Blockade. Cell Rep. 18, 248–262 (2017).

# Field-specific reporting

Please select the one below that is the best fit for your research. If you are not sure, read the appropriate sections before making your selection.

☒ Life sciences  ☐ Behavioural & social sciences  ☐ Ecological, evolutionary & environmental sciences

For a reference copy of the document with all sections, see nature.com/documents/nr-reporting-summary-flat.pdf

# Life sciences study design

All studies must disclose on these points even when the disclosure is negative.

| | |
|---|---|
| Sample size | 180 women with early and locally advanced breast cancer planned to undergo neoadjuvant treatment were prospectively enrolled the molecular profiling study described (TransNEO). Of these, 12 were excluded and not sequenced (reasons: no research biopsy taken (n=6), co-diagnosis of metastatic disease (n=3), recruited to early stage clinical trials (n=2), died early during therapy (n=1)). Tumours from the 168 remaining women were molecularly profiled, of which 155 had associations with RCB and received adequate therapy exposure (defined as more than 1 cycle of chemotherapy and, if HER2+, more than 1 cycle of targeted therapy). This is summarised in Extended Data Figure 1 and in the Methods section.<br><br>For the validation dataset, sequenced cases within the control arm of the ARTemis trial (n=38) and cases within the PBCP study (n=37) that received neoadjuvant therapy and had DNA, RNA, and digital pathology data were used for validation (summarised in Extended Data Figure 1). |
| Data exclusions | To determine associations between response, only cases which had molecular/digital pathology data and received more than 1 cycle of chemotherapy and, if HER2+, received more than one cycle of targeted therapy were included (n=155 as described in Extended Data Figure 1 and Methods). These exclusion criteria were pre-established prior to commencing analysis to ensure that associations with response were only derived using data from patients treated with adequate therapy exposure (defined as more than one cycle of therapy). |
| Replication | The findings were validated in an independent dataset comprising 75 cases with DNA, RNA and digital pathology data. |
| Randomization | Randomization not applicable - all cases were treated with standard of care therapy regimens. |
| Blinding | Blinding not applicable - no group allocations. |

# Reporting for specific materials, systems and methods

We require information from authors about some types of materials, experimental systems and methods used in many studies. Here, indicate whether each material, system or method listed is relevant to your study. If you are not sure if a list item applies to your research, read the appropriate section before selecting a response.

### Materials & experimental systems

| n/a | Involved in the study |
|---|---|
| ☒ | ☐ Antibodies |
| ☒ | ☐ Eukaryotic cell lines |
| ☒ | ☐ Palaeontology and archaeology |
| ☒ | ☐ Animals and other organisms |
| ☐ | ☒ Human research participants |
| ☐ | ☒ Clinical data |
| ☒ | ☐ Dual use research of concern |

### Methods

| n/a | Involved in the study |
|---|---|
| ☒ | ☐ ChIP-seq |
| ☒ | ☐ Flow cytometry |
| ☒ | ☐ MRI-based neuroimaging |

# Human research participants

| | |
|---|---|
| Population characteristics | All participants within the TransNEO and PBCP studies were women diagnosed with early/locally advanced breast cancer and treated with neoadjuvant chemotherapy (and anti-HER2 therapy if HER2+) between 2013-2018. Participant characteristics are included within Supplementary data table 1.<br><br>The population characteristics of the patients used in the control arm of the ARTemis Study are described in Earl, H. M. et al. Efficacy of neoadjuvant bevacizumab added to docetaxel followed by fluorouracil, epirubicin, and cyclophosphamide, for women with HER2-negative early breast cancer (ARTemis): an open-label, randomised, phase 3 trial. Lancet. Oncol. 16, 656–66 (2015). Link to article: https://doi.org/10.1016/S1470-2045(15)70137-3 |
| Recruitment | Within the TransNEO and PBCP studies, all women with early/locally advanced breast cancer presenting to Cambridge University Hospitals NHS Foundation Trust and planned to undergo pre-operative chemotherapy were approached by the Cambridge Breast Cancer Unit research team and offered participation within the study.<br><br>Inclusion criteria included:<br>1. Patient with histological diagnosis of invasive breast cancer<br>2. Patient receiving neoadjuvant therapy (chemotherapy and/or hormonal therapy)<br>3. Able to give informed consent<br>4. ECOG 0-2<br><br>In the ARTemis trial, key inclusion and exclusion criteria are available at https://www.clinicaltrialsregister.eu/ctr-search/search?query=2008-002322-11 and the trial description and results have been previously published https://doi.org/10.1016/S1470-2045(15)70137-3<br><br>There is no selection bias within this study: any patient identified in standard of care clinical practice to benefit from neoadjuvant therapy was approached to take part in the study, and all those who consented and donated tumour tissue were included in the study if they received more than one cycle of therapy and response assessment was available post therapy (Extended data figure 1). |
| Ethics oversight | East of England Research Ethics Committee: 12/EE/0484 (TransNEO), 18/EE/0251 (PBCP)<br>South East Research Ethics Committee: 08/H1102/104 (ARTemis) |

Note that full information on the approval of the study protocol must also be provided in the manuscript.

# Clinical data

| | |
|---|---|
| Clinical trial registration | ARTemis clinical trial: EudraCT Number 2008-002322-11, UK South East REC Number: 08/H1102/104, https://www.clinicaltrialsregister.eu/ctr-search/search?query=2008-002322-11 |
| Study protocol | ARTemis clinical trial protocols:<br>https://www.clinicaltrialsregister.eu/ctr-search/trial/2008-002322-11/GB<br>https://warwick.ac.uk/fac/sci/med/research/ctu/trials/cancer/artemis/ |
| Data collection | The ARTemis clinical trial collected data recruited women with early invasive breast cancer (radiological tumour size >20 mm, with or without axillary involvement), at 66 centres in the UK between May 7, 2009, and Jan 9, 2013. Full details of the trial have been published and are available within the supplementary material of the trial publication in Lancet Oncology: https://doi.org/10.1016/S1470-2045(15)70137-3 |
| Outcomes | In the ARTemis trial, the primary endpoint was defined as complete pathological response rates after neo-adjuvant chemotherapy defined as no residual invasive carcinoma within the breast (DCIS permitted) AND no evidence of metastatic disease within the lymph nodes. The secondary endpoints were:<br>1. Disease-Free Survival<br>2. Overall Survival<br>3. Complete pathological response rates rate in the breast alone<br>4. Radiological (ultrasound) response after 3 and after 6 cycles of chemotherapy. Rate of breast conservation<br>Toxicities, including in particular cardiac safety and surgical complications (wound healing, bleeding, and thrombosis).<br><br>The results and assessment of these endpoints have already been published in Lancet Oncology: https://doi.org/10.1016/S1470-2045(15)70137-3 |

