## [Peer Review File · Nature]

Manuscript Title: Multi-omic machine learning predictor of breast cancer therapy response

Reviewer Comments & Author Rebuttals

Reviewer Reports on the Initial Version:

Referee #1 (Remarks to the Author):

In the current manuscript, Sammut et al performed a multiomics analyses of breast cancer samples obtained before treatment, and correlated to sensitivity to chemotherapy. They further integrated the data to develop a multiomics predictor of response to neoadjuvant chemotherapy. This is a very large amount of work. The text and the figures are pleasant to read. I have some major comments about the interpretation of the data.

1. While pCR is validated as a surrogate of outcome in TNBC and Her2+ BC, its relevance is more controversial in patients with HR+ BC. It's therefore unclear whether the predictor will have any clinical utility. The authors should test its value to predict relapse
2. The paper is expected to show that multi-omics data integration improves AUC. Nevertheless, the paper does not provide evidence that it's the integration of multi-omics data itself that improves performance. Indeed, some simple, previously reported molecular characteristics (TP53, TILs, ESR1 expression, proliferation modules), explain in large part the AUC increase and do not require data integration and machine Learning. To address this comment, the authors should test the AUC when some standard biological parameters are added to the clinical variables. As example; proliferation genes, ESR1, PGR, ERBB2 expression + TIL + TP53 mutations. Authors should also test how PAM50 improves AUC.
The need for multi-omics data to build the predictor is also challenged by the Figure 4d that clearly shows that RNA (and probably ESR1, ERBB2, proliferation, T cell markers) + clinical variable allows reaching an AUC>0.8
3. Some findings are not extremely robust to enter in the data integration / ML and do not have major importance in the final predictor. The reviewer encourages the authors to add one more layer of validation before the external validation, to test that some findings are robust enough to enter in the data integration step.
4. Clinical phenotypes have a very limited predictive power. The authors could maybe add a Ki67 staining in the model, since it's a standard marker in translational research performed in neoadjuvant setting. Also, as mentioned before, it would be important to know the value of clinical + PAM50 since 1st generation genomic signatures are increasingly used in BC. The lack of significance for cT and Her2 is surprising because these are two major drivers of pCR. This could maybe mean that the FDR threshold is too stringent for clinical parameters. In general, the reviewer would recommend the authors to use already published clinical algorithms rather than developing a new one based on small number of patients.
5. lines 142-184 : since most of these parameters highly correlate with ER status, it's important to show that their predictive value is independent to ER, or the authors just report a surrogate of ER status
6. Extended sup Fig 9b is extremely important but quite difficult to understand. I would suggest this figure moves to the main set of figures and that authors make it more simple to read (as example, LR, RF, SVC should be defined in the figure legend).

7. Figure 1 : The flow between step 1 and 2 unclear if the reader starts the paper by reading figure 1. Do the authors integrate only the factors significant in univariate ? or do they integrate all parameters ?

8. Figure 4e : the goal of neoadj therapy is to allow breast conserving surgery, so there is no reason to deny NAT if a patient is predicted to have clinical response without pCR.

9. Variables related to therapies should be removed from the model since they can't be used as a predictor in a prospective setting

Referee #2 (Remarks to the Author):

A. Summary of key results.

This interesting study uses a multi-platform and multi-OMICS approach to predict complete pathologic response at surgery in a cohort of breast cancer patients who receive neo-adjuvant therapy. Data from clinical and pathological records, DNA, RNA and digital pathology were combined using an ensemble of machine learning algorithms to predict pathologic response. The results were confirmed in two test cohorts.

B. Originality and significance.

The study takes advantage of a novel dataset of cases that were enrolled for generation of original data. The significance of the study is high because it shows how high dimensionality data can be processed and integrated in an interpretable and logical way that will allow a future clinical implementation. Measurements can be generated in a CLIA clinical laboratory and the fixed algorithm used for clinical assay development.

A small weakness is that the study generated many confirmatory results, but only a few novel insights and no conceptually novel mechanisms related to our understanding of treatment response/resistance in breast cancer. As such, the relatively small number of cases in the discovery and validation cohort is sufficient, since no new biological concepts need to be validated.

C. Data & methodology: validity of approach, quality of data, quality of presentation Strengths:

- This is a nicely designed and elegant study because of its simplicity.
- A broad range of data are aggregated in an interpretable fashion.
- The discovery cohort, which consists of 168 cases with frozen tissue biopsies available, is well balanced in terms of breast cancer subtype and response to treatment endpoint
- The choice of variables (features) that were analyzed is based on a sound rationale and uses well established methodologies for both, data generation and data analysis.
- The ensemble approach is quite interesting and innovative and the variable importance highlights which of the features are most informative in the outcomes prediction.
- The systematic analysis of the impact of clinical, DNA, RNA and digital pathology features in the outcomes prediction can be generalized across cancer types and data sources.

Weaknesses:

- Cohort sizes are relatively small and it is unclear whether all the features used in the model can be obtained from FFPE tissues. This would be important to mention because of its clinical relevance.
- Please include the rank list of features in the supplementary data to complement the color-based figures.
- It would be helpful to see the performance of an ensemble model that only includes the highest ranking features shown in Figure 4b and that can be applied to FFPE tissues.

D. Appropriate use of statistics and treatment of uncertainties

Please refer to comments from statistical reviewers

E. Conclusions: robustness, validity, reliability

The conclusions from the data are valid. The machine learning models were developed using a cross-validation and bootstrapping approach and individual models were fixed before testing. The ensemble approach increases the robustness. I would be helpful to understand the difference between algorithms in the outcomes prediction and to address cases that reveal a large difference in prediction results.

F. Suggested improvements: experiments, data for possible revision
Please see suggestions in section C.

G. References: appropriate credit to previous work?

The reference section is adequate.

H. Clarity and context: lucidity of abstract/summary, appropriateness of abstract, introduction and conclusions

The manuscript is well written and balanced. It should be easy to understand by reader from different disciplines. The expanded data and methods sections contain enough depth to allow evaluation by subject matter experts.

Referee #3 (Remarks to the Author):

This submission from Sammut et al. details the development, training, and independent validation of a multi-omic machine learning (ML) predictor of neoadjuvant therapeutic response across the major subtypes of breast cancer (ER+, HER2+, triple negative). The key finding is that the fully integrated model that incorporates clinical/pathological, DNA, RNA, digital pathology, and treatment data yielded the best performance (precision-recall AUC=0.87 in the validation set from the ARTemis trial), although the importance of treatment data was not weighted as highly by the model. Other important results include the finding that although the training set considered response as binary (pathological complete response (pCR) versus residual disease), the fully integrated model could subdivide residual disease into three categories of residual cancer burden (RCB).

Originality and significance are deemed high. ML prediction of neoadjuvant therapeutic response has long needed a rigorous incorporation of tumor microenvironment variables, including (and perhaps especially) the immune response, and to do so here through precision-recall approaches to account for imbalanced observations is a strength. An additional strength is the inclusion of an ensemble classification approach. The work is of excellent quality, results are clearly explained, the manuscript is well-written, and for the most part the study is statically robust.

Three points for the authors which, if addressed, would improve understanding and perhaps uptake of this ML predictor are as follows. One, it is a little perplexing that treatment parameters/features appear to be less important to the overall integrated training model (Figure 4b, Extended Data Figure 9b), as they are omitted from the description of the results on manuscript page 12. Two, the importance and signed importance z-scores for model features in these same figures suggest that clinical/pathological data such as HER2 status, ER status, and histological grade also contribute relatively little to overall classification, although this varies between individual models. Three, while the overall classifier takes an average of the three algorithms, across the board the random forest approach appears to discount immune features. What is the reason for this?

Referee #4 (Remarks to the Author):

The authors collected clinical, digital pathology, genomic and transcriptomic profiles pre-treatment biopsies of breast tumors from 168 patients treated with chemotherapy +/-HER-targeted therapy prior to surgery. First, they performed statistical analysis to identify features associated with pathological complete response (pCR) using each modality separately. Then, they developed machine learning models to predict pathological complete response (pCR) using all modalities. They also validated their results using an independent dataset. The statistical and machine learning methods used seem sound. Overall, the manuscript is clear and accessible. This work would be of interest to breast cancer research community. Novelty of results are outside the scope of my expertise, and I was unable to assess fully.

Specific comments

-Please define the abbreviation the first time you use it in the text (e.g., OR, CI, FDR line 135) -When tested using the external cohort, models built with clinical+RNA, clinical+DNA+RNA, clinical+DNA+RNA+digital pathology and fully integrated didn't show significant performance difference. Can authors comment on this in discussion section? - <https://github.com/cclab-339brca/neoadjuvant-therapy-response-predictor> -> The link was broken. I couldn't review.

-Figures:

--All error bars and test statistics are not defined in several corresponding figure legends. --Fig 4d what does dotted line represent? Marker for training is missing.

Author Rebuttals to Initial Comments:

Referee 1:

In the current manuscript, Sammut et al performed a multiomics analyses of breast cancer samples obtained before treatment and correlated to sensitivity to chemotherapy. They further integrated the data to develop a multiomics predictor of response to neoadjuvant chemotherapy. This is a very large amount of work. The text and the figures are pleasant to read. I have some major comments about the interpretation of the data.

We would like to thank the referee for taking the time to go through our manuscript and for their positive and thorough comments. We are pleased this referee found the text and figures pleasant to read and also recognizes the large amount of work we have done. Indeed, this is the largest neoadjuvant breast cancer series that integrates pre-therapy shallow whole genome sequencing, deep whole exome sequencing, whole transcriptome and digital pathology data and correlates these with a spectrum of post-therapy tumour response.

1. While pCR is validated as a surrogate of outcome in TNBC and Her2+ BC, its relevance is more controversial in patients with HR+ BC. It's therefore unclear whether the predictor will have any clinical utility. The authors should test its value to predict relapse

Clinical Utility: We agree that the pCR/RCB classification has been previously validated as a surrogate of long-term survival, with the strongest association observed in ER-HER2-(TNBC) and HER2+ breast cancer¹. However, since the submission of this manuscript, a further manuscript that we have co-authored has been accepted in Lancet Oncology (Yau et al, Lancet Oncology, in press), which performed the largest RCB meta-analysis to date (5,161 patients from 12 international cancer centres), to definitively show that both pCR and the RCB classification are strongly predictive of relapse free survival across the full spectrum of the disease, including in ER+HER2-. We have provided a copy of the manuscript for reference (we would like to ask this is kept strictly confidential until Lancet Oncology embargo is over).

In our current manuscript we have shown that the multi-omic machine learning predictor we developed:

1. Predicts pCR prior to commencing therapy (which is strongly predictive of relapse free survival as per the Lancet Oncology paper).
2. Predicts chemoresistance prior to commencing therapy, as it statistically correlates with the degree of residual disease post-therapy (Extended Fig 10). RCB is strongly predictive of relapse free survival across the spectrum of breast cancer, including ER+HER2- cases.

Given these two observations, the potential clinical utility lies on the multi-omic predictor's ability to direct therapy selection in:

1. Patients with chemosensitive tumours who will respond well to standard cytotoxic and targeted therapies and should receive current protocol regimens as they are more likely to have breast conserving surgery² and their prognosis (relapse free survival) is excellent¹.
2. Patients with chemoresistant tumours who are unlikely to respond to standard cytotoxic therapies and have a poorer prognosis (relapse free survival)¹ and lower rates of breast conserving surgery². These patients should be identified early, and instead directed towards neoadjuvant clinical trials as standard therapies are not as effective. Arguably, these are the patients that benefit most from participation in clinical trials.

Action taken: This comment is related to comment 8 below, and we have addressed both comments by amending the text on page 13 lines 328-331 as well as amending Figure 4e.

In a clinical workflow the predictive models could be applied to patients who are candidates for neoadjuvant therapy: any predicted to have chemoresistant tumours should be considered for enrolment into investigational neoadjuvant clinical trials of novel therapies, as their prognosis is likely to be poor if they are treated with standard of care therapies (Fig. 4e).

Figure 4e: Amended to address Reviewer 1's comments (1 and 8)

Predicting relapse: The model we describe was specifically trained to predict response and not relapse. We have not undertaken an analysis of relapse as the median follow-up across all datasets is short (four years) and few patients have relapsed within that time frame (n=21). As ER+ tumours often relapse beyond five years, a survival analysis on immature follow-up data would not accurately capture the dynamics of relapse in ER+ disease³. We would like to highlight nevertheless that, of the 21 patients who relapsed, 20 had residual disease post-chemotherapy, in keeping with pCR being a strong predictor of outcome (p=0.02 Fisher's Exact Test).

2. The paper is expected to show that multi-omics data integration improves AUC. Nevertheless, the paper does not provide evidence that it's the integration of multi-omics data itself that improves performance. Indeed, some simple, previously reported molecular characteristics (TP53, TILs, ESR1 expression, proliferation modules), explain in large part the AUC increase and do not require data integration and machine Learning. To address this comment, the authors should test the AUC when some standard biological parameters are added to the clinical variables. As example; proliferation genes, ESR1, PGR, ERBB2 expression + TIL + TP53 mutations. Authors should also test how PAM50 improves AUC. The need for multi-omics data to build the predictor is also challenged by the Figure 4d that clearly shows that RNA (and probably ESR1, ERBB2, proliferation, T cell markers) + clinical variable allows reaching an AUC>0.8

Multi-omic data point 1: The reviewer highlights a very important point. When developing multi-omic prediction models it is important to show, in a data driven way, that integration improves the discriminatory power of a classifier.

In this manuscript we show that both the quantification of biologically relevant features, as well as their integration, are key to the performance of our predictor. We disagree that multi-omic features are not required. Indeed, the features which the referee points out that explain the large part in the AUC increase are obtained from multi-omic data: *TP53* (DNA), TILs (digital pathology), *ESR1* expression and proliferation (RNA).

Crucially, given the large number of features available, testing 'ad-hoc' combinations is not a statistically robust approach: the process of testing one combination after the other to find the optimal model would lead to multiple-testing errors. The machine learning framework is required because it combines/integrates these biologically relevant features by identifying the optimum combination of features and their weights (coefficients) to predict response.

Rather than adding biological parameters 'ad hoc' to the clinical model, we present a systematic approach in which layers of data categorised by profiling modality are added to the baseline clinical model (+DNA, +RNA, etc) as shown in Figure 4d. This unbiased approach is statistically robust and objective. It is a data-driven approach with robust statistical significance within the training dataset and then rigorous external validation. This is the accepted and universally recommended way of developing a new predictor. By using an ensemble machine learning approach, the algorithms identified the optimum multivariable combinations that result in the best prediction. The description of 'standard' biological features is very subjective, especially when it comes to molecular data, which is why we opted to use a more objective

data-driven approach to model generation that is not influenced by human bias and multiple testing problems. Additionally, because of the approach used the reader can easily appreciate which interaction of features provides the greatest contributions to response. We feel that is best exemplified by Extended Figure 9b, which highlights the benefit of this approach, as the Reviewer mentions in comment 6. Indeed, this figure (along with Figure 4b) has allowed the referee to quickly check which variables are the most important in comparison to the rest and correlate these with the biology of the disease.

In summary, machine learning provides the ideal framework to derive a multivariable model in which the most important features are found and combined in an unbiased, data-driven way. The main challenge for machine learning is to ensure that the derived multivariable model is robust. The gold standard to demonstrate robustness and reliability is to freeze the model after training and validate it on an external, independent dataset. This is exactly the procedure that we followed, successfully validating the performance of the model on an independent dataset.

PAM50: There is very little heterogeneity between clinical ER/HER2 status and PAM50 subtypes, with the majority of ER+HER2- tumours being luminal A and B and the majority of ER-HER2- tumours being basal-like. We tested if the addition of the PAM50 subtypes (RNA) to our full model (logistic regression for illustration) would increase its predictive ability. The likelihood ratio test showed that the PAM50 classification did not add value to the model ($p=0.2$). We have chosen not to include PAM50 (RNA) as we feel that the biology of the disease is better modelled by more granular RNA features. Furthermore, the commercial Prosigna test (PAM50) is not routinely used in most centres and is only indicated for post-operative (not neoadjuvant) therapy decisions in post-menopausal women with ER+ early-stage breast cancer.

Multi-omic data point 2: Indeed, as the reviewer rightfully notes the greatest increase in AUC occurs when RNA features are added to the (training) model. However, the performance of the model in the training dataset also clearly shows an increase in AUC as successive levels of multimodal data are added. We do not observe the same magnitude of increase in the validation set as it is smaller in sample size. Furthermore, it is not surprising to see that some features contribute more than others: for example, the RNA-seq data comprises features from both the cell-autonomous compartment and the tumour microenvironment, as well as their functional states, whilst the DNA data predominantly captures tumour-specific features. Additionally, Extended Figure 9b shows that the three machine learning classifiers use features from all modalities to generate a prediction. While the greatest increase is contributed after adding RNA features, the work we have done shows that there is benefit in adding

incremental levels of data, with the fully integrated model using data from all modalities, rather than just one modality.

Action taken: We have added the following sentence on page 12, lines 319 - 320: *The fully integrated model relied on features obtained from all modalities of data, with RNA features having the largest contribution (Fig. 4b, Extended Data Fig. 9b).*

3. Some findings are not extremely robust to enter in the data integration / ML and do not have major importance in the final predictor. The reviewer encourages the authors to add one more layer of validation before the external validation, to test that some findings are robust enough to enter in the data integration step.

We agree with the reviewer that it is important to make sure that the feature selection process is made clear in the manuscript. The molecular and digital pathology features used to train the machine learning model were selected due to being significantly associated with response in the training dataset on a univariable level. We then adopted a standard machine learning approach where subsequent feature selection was done in an unbiased fashion by the pipeline with collinearity removal and k-best feature selection, with all hyperparameters optimised using 5-fold cross-validation in the training set to maximise the AUC-ROC as described within the Methods. This allowed the three classification pipelines to independently select which feature combinations best predict response and optimise feature weightings, and as a result greatly decreased the risk of model overfitting. The fact that some features contribute less to the overall model is reassuring, as it shows that the machine learning pipeline is performing as expected. The integration of multiple features and the explicit modelling of interactions (for example, in the Random Forest) between features are the reasons why some features that look promising in univariable models are down weighted by the final model, highlighting the value of data integration.

The intermediate layer of validation step that the referee suggests is naturally provided by the cross-validation procedure. Moreover, the model validates very well in external datasets, which is the ultimate gold standard of validation, so we disagree that an additional layer of validation needs to be added.

4. Clinical phenotypes have a very limited predictive power. The authors could maybe add a Ki67 staining in the model, since it's a standard marker in translational research performed in neoadjuvant setting. Also, as mentioned before, it would be important to know the value of clinical + PAM50 since 1st generation genomic signatures are increasingly used in BC. The lack of significance for cT and Her2 is surprising because these are two major drivers of pCR. This could maybe mean that the FDR threshold is too stringent for clinical parameters. In general, the reviewer would recommend the authors to use already published clinical algorithms rather than developing a new one based on small number of patients.

We agree with the reviewer that clinical phenotypes have limited predictive power.

Ki67: We did not include Ki67 IHC staining in the model as: (1) it is not currently routinely performed in the majority of labs in the UK and (2) it is not currently recommended in the US (NCCN, ASCO/CAP), UK (RCPATH) and international guidelines because of a lack of consensus on scoring, definition of low versus high expression and an appropriate cut point for positivity (International Collaboration on Cancer reporting guidelines: <http://www.iccr-cancer.org/getattachment/Datasets/Published-Datasets/Breast/Invasive-Carcinoma-of-the-Breast/ICCR-Invasive-breast-1st-edn-v1-1-bookmark.pdf>, page 28). Working groups that advocate the use of Ki67 also state in their guidelines that it is not accurate between 5% and 30% (where the majority of ER positive cancers fall)⁴. Ki67 staining was not routinely performed in the NHS and tissue availability would make it impossible to obtain it now. The model, however, incorporates the Genomic Grade Index⁵ (RNA), which has been shown to correlate well with Ki67 IHC status and has a better performance as an outcome predictor⁶.

PAM50: As we discussed in comment 2, the addition of more granular features derived from RNA data (rather than the categorical RNA-based PAM50 classification) is favoured by the full model. We have also shown that PAM50 did not add discrimination power to the overall predictor. We would like to stress that the intrinsic subtypes were not generated as a prognostic tool but were the result of an unsupervised cluster analysis of gene expression data. In addition, the commercial assay (Prosigna) is not universally used and does not have a licence for use in neoadjuvant therapy decisions. It is worth noting that, as we are releasing our code and all the raw data, readers can build models using permutations of different commercial assays if they wish to do so (eg Prosigna, OncotypeDX, etc).

Development of clinical model: In our pre-FDR corrected logistic regression model (Extended Data Figure 2c), the association with clinical tumour size (cT) was borderline significant ($p=0.04$), though following FDR correction this increased to 0.1, which was above

our pre-specified significance cut-off (<0.05). Regarding the performance of our clinical model: the AUC obtained was 0.77 in training and 0.7 in validation, which is comparable to the AUCs of clinical models developed from larger series, which we have summarised in Table R1.1.

Table R1.1 Reported AUCs for published neoadjuvant clinical nomograms

Study	Number of cases	Predictor AUC
Rouzier et al, 2005 ⁷	496	0.77
Lee et al, 2010 ⁸	100	0.72
Jin et al, 2016 ⁹	815	0.70
Pu et al, 2020 ¹⁰	165	0.76

As a comparison, we assessed the performance of the NHS Predict calculator¹¹ (<https://breast.predict.nhs.uk/>), which uses established clinical variables to predict survival in the adjuvant (rather than neoadjuvant) setting (Figure R1.1). While the scope of this calculator is very different from the clinical predictor we use, it is worth noting that the AUCs obtained were slightly lower than those obtained by our predictor.

Figure R1.1 Performance of the NHS Predict algorithm across discovery and validation datasets.

The fact that all clinical models provide a similar performance level, including the one derived using our own framework, suggests that the bottleneck is indeed in the clinical features themselves and not the statistical methods. As a consequence, we could potentially use any of these models as a benchmark. We have decided to use the one derived using our machine learning framework as that way we ensure that the improvements observed after integration are purely due to the addition of new features and not caused by the use of a different statistical framework.

Action taken: We have added the following sentence at page 12 lines 310 - 312 to reflect this: “The baseline clinical model as implemented using our machine learning algorithms performed similarly to other clinical predictors reported in larger datasets^{8,9}”.

5. lines 142-184 : since most of these parameters highly correlate with ER status, it's important to show that their predictive value is independent to ER, or the authors just report a surrogate of ER status

Biological features associated with response are often correlated with ER and HER2 status (eg: proliferation is associated with response in ER+HER2- and ER-HER2- tumours, immune activation in ER+HER2- and HER2+ tumours). We show this clearly in a feature correlation heatmap (Extended Data Figure 9a). However, the association of a biological feature with ER/HER2 status does not make it any less important.

We must stress that the analyses we have performed in that section aimed to identify candidate features that might be relevant because of their association to pCR. Furthermore, the final predictor model considers the effect of ER and shows that the rest of the features have a predictive ability that is independent of ER. As shown in Extended Data Figure 9b, within the integrated model, the machine learning algorithms still rely heavily on the contribution of these 'surrogate' features, with the random forest particularly preferring other metrics than HER2 expression.

However, we have looked at the genomic correlations in our univariable analysis from lines 142-184 and show that in most cases the significance we observe is maintained even when ER is added to the model, as shown in Table R1.2

Table R1.2 Genomic features associated with response in a logistic model

	PIK3CA	TP53	TMB	Subclonal TMB	Neoantigens	HRD	Chromosomal instability
ER p	0.00054	0.0037	0.00023	0.0008	0.00022	0.0031	0.013
Feature p	0.17	0.21	0.021	0.0095	0.17	0.0027	0.033

We chose to describe the univariable associations with genomic landscape across HER2 status as: (a) we show in the RNA analysis that metagenes of proliferation are associated with response in HER2- but not HER2+ tumours (Extended Data Figure 6) and (b) we subsequently show that RNA proliferation metagenes are associated with features related to DNA genomic instability (Extended Data Figure 9a). We hoped that the manuscript would be more accessible to a less expert audience if we presented response across the genomic landscape by stratifying by HER2 status, thereby allowing the reader to rapidly make an association between the DNA mutational and RNA proliferative landscape.

6. Extended sup Fig 9b is extremely important but quite difficult to understand. I would suggest this figure moves to the main set of figures and that authors make it more simple to read (as example, LR, RF, SVC should be defined in the figure legend).

We agree that this supplementary figure conveys a very important point. We placed this figure in the supplementary section as we feel that it would predominantly be of interest to a specialist readership and chose to present the summary version as a main figure instead (Figure 4b), which we feel is more accessible to a broader audience. Both related figures (Figure 4b and Extended data Figure 9b) are always referred to in tandem in the text.

Action taken: We have removed all abbreviations within Extended data Figure 9b.

7. Figure 1: The flow between step 1 and 2 unclear if the reader starts the paper by reading figure 1. Do the authors integrate only the factors significant in univariate ? or do they integrate all parameters ?

We have generated an updated version of Figure 1 to show the reader that we integrated the molecular and digital pathology features found associated with response to neoadjuvant therapy in the first part of the manuscript.

Action taken: We have altered the title of Step 2 from: “Step 2: Data integration using a multi-omic machine learning model” to: “Step 2: Data integration of features identified in Step 1 using a multi-omic machine learning model” to ensure that there is a clear connection between the discovery of associations in step 1, and their subsequent integration in step 2. We have also amended the corresponding figure legend as shown below.

Revised Figure 1 and figure legend: Overview of the study design. Pre-therapy breast tumours from 168 patients were profiled using DNA and RNA sequencing and digital pathology analysis. Response was

assessed on completion of neoadjuvant therapy using the Residual Cancer Burden (RCB) classification. Individual pre-therapy clinical, molecular and digital pathology features associated with pathological complete response (pCR) were identified (step 1) and then integrated within machine learning models to predict response, which were then validated in an independent dataset (step 2).

8. Figure 4e: the goal of neoadj therapy is to allow breast conserving surgery, so there is no reason to deny NAT if a patient is predicted to have clinical response without pCR.

We concur that neoadjuvant therapy increases the rate of breast conserving surgery but in addition it also has significant prognostic value as discussed in comment 1 (Yau et al, Lancet Oncology, in press).

Action taken: We have addressed this point in comment 1 and altered Figure 4e and rephrased the corresponding main text, as follows (page 13 lines 328-331):

In a clinical workflow the predictive models could be applied to patients who are candidates for neoadjuvant therapy: any predicted to have chemoresistant tumours should be considered for enrolment into investigational neoadjuvant clinical trials of novel therapies, as their prognosis is likely to be poor if they are treated with standard of care therapies (Fig. 4e).

Figure 4e: Amended to address Reviewer 1's comments (1 and 8)

9. Variables related to therapies should be removed from the model since they can't be used as a predictor in a prospective setting

We trained a data integration model that does not include therapy features, in addition to the one that does (Figure 4d). Chemotherapy sequence (taxane>anthracycline vs anthracycline>taxane vs anthracycline-free regimens) and number of chemotherapy cycles were used to train the framework to see whether treatment effects improve model AUC. We

have included these variables to show the reader that the framework can also model treatment effects, and is in keeping with other available predictors, which allow the user to specify chemotherapy backbone (eg the NHS Predict calculator¹¹). We note that the treatment type that the patient is going to receive is commonly known at the moment of the prediction, and therefore it can be used in a prospective setting. We also note that other referees considered that the inclusion of treatment effects was a strength of the model.

Referee 2:

A. Summary of key results.

This interesting study uses a multi-platform and multi-OMICS approach to predict complete pathologic response at surgery in a cohort of breast cancer patients who receive neo-adjuvant therapy. Data from clinical and pathological records, DNA, RNA and digital pathology were combined using an ensemble of machine learning algorithms to predict pathologic response. The results were confirmed in two test cohorts.

B. Originality and significance.

The study takes advantage of a novel dataset of cases that were enrolled for generation of original data. The significance of the study is high because it shows how high dimensionality data can be processed and integrated in an interpretable and logical way that will allow a future clinical implementation. Measurements can be generated in a CLIA clinical laboratory and the fixed algorithm used for clinical assay development.

We thank the referee for their very positive comments, and for acknowledging the importance of this translational study and for being so complimentary on our approach. We note these comments: “...it shows how high dimensionality data can be processed and integrated in an interpretable and logical way...” and “...the fixed algorithm used for clinical assay development...”.

A small weakness is that the study generated many confirmatory results, but only a few novel insights and no conceptually novel mechanisms related to our understanding of treatment response/resistance in breast cancer. As such, the relatively small number of cases in the discovery and validation cohort is sufficient, since no new biological concepts need to be validated.

We were pleased to see that known biology was recapitulated within this study, as it shows that the dataset we generated is truly representative of the disease, and that the analyses we report in the manuscript are robust. We believe that this study highlights the following novel discoveries and concepts:

1. First to show that most predictive genomic, transcriptomic and digital pathology features are monotonically associated with the degree of residual disease post therapy: as the

features decrease/increase in enrichment, so does the degree of residual disease. This is a novel observation as it shows that the degree of response is determined by biological feature abundance within the pre-therapy landscape.

2. First to show how the close interplay between proliferation and immune activation is associated with the degree of residual disease post therapy (Figure 3e and validated in external datasets: Extended Data Figure 7f).
3. First to show the association between LOH HLA and response prediction (Charlie Swanton's group had shown HLA LOH was prognostic in lung cancer¹²) and map the contribution of the tumour ecosystem to response by combining mechanisms of immune exclusion and dysfunction (and we note this is orthogonally validated by lymphocytic infiltration using digital pathology).
4. Also showed how a systematic discovery of biological features associated with response derived from high dimensional multimodal data can be integrated using ensemble machine learning to create predictors of response. This overall data analysis and integration pipeline is readily portable to other tumour types and is adaptable to the available data in any given study.
5. From a clinical point of view, the model could be used to identify which patients are unlikely to respond to standard cytotoxic and targeted therapies, and instead directed towards clinical trials, as arguably they are the ones that stand to benefit the most from them.

C. Data & methodology: validity of approach, quality of data, quality of presentation Strengths:

- This is a nicely designed and elegant study because of its simplicity.
- A broad range of data are aggregated in an interpretable fashion.
- The discovery cohort, which consists of 168 cases with frozen tissue biopsies available, is well balanced in terms of breast cancer subtype and response to treatment endpoint
- The choice of variables (features) that were analyzed is based on a sound rationale and uses well established methodologies for both, data generation and data analysis.
- The ensemble approach is quite interesting and innovative and the variable importance highlights which of the features are most informative in the outcomes prediction.
- The systematic analysis of the impact of clinical, DNA, RNA and digital pathology features in the outcomes prediction can be generalized across cancer types and data sources.

We are grateful the reviewer shares our enthusiasm for this work and finds our approach interesting, sound, and innovative.

Weaknesses:

- Cohort sizes are relatively small and it is unclear whether all the features used in the model can be obtained from FFPE tissues. This would be important to mention because of its clinical relevance.

Cohort size: This is the largest neoadjuvant breast cancer study we know of that integrates pre-therapy sWGS, deep whole exome, whole transcriptome and digital pathology data with a spectrum of response to therapy quantified at surgery.

FFPE tissues: Although generating genome and transcriptome data from FFPE tissues has historically been challenging, there have been substantial improvements in FFPE library preparation protocols and analytical methods within the past few years, with studies now reporting high concordance for both DNA and RNA measurements between fresh frozen and matched FFPE tumour cores^{13,14}. Additionally, we, along with others, have shown that the shorter the time period between generating FFPE tumour material and its analysis, the better the quality of the DNA/RNA and the higher the correlation with fresh frozen tumours¹⁵. The optimization of assays for FFPE and derivation of an FFPE-adjusted predictor using an ensemble ML approach as we described here is beyond the scope of our current manuscript.

RNA: All the RNA-seq features obtained from fresh frozen tumours can be obtained from FFPE tumours: we have explored this by analysing an independent set of 31 breast cancer cases for which we have matched RNA sequencing data from both an FFPE core and fresh tissue core (unpublished data). As shown in similar studies¹³, the median whole transcriptome correlation between related samples was 0.89 (range: 0.77 – 0.98). We computed the correlation of all the RNA features used by the predictor in fresh frozen and the matched FFPE core and found a high correlation across all features (R: 0.44 – 0.75). This gives us confidence that the RNA features required by the predictor can reliably and reproducibly be extracted from FFPE tissues.

DNA: We, and others, have also previously shown that the DNA features used by the model, including copy number estimation, loss of heterozygosity and mutational landscape characterisation, can be readily obtained from FFPE tumours^{14–16}. We have used Mutect2 to call mutations in this study: this variant caller has an orientation-bias filter that is specifically designed to filter somatic variant calls for sequence context-dependent artifacts including deamination, and is therefore well suited to FFPE mutation calling.

Digital pathology: Lymphocyte density as obtained through a digital pathology analysis of FFPE slides has already been shown to be associated with pCR^{17,18}. We have used the same digital pathology algorithms in our fresh frozen tumour cohort that were used in the FFPE cohort (as described in the Methods section).

Given these observations, we are confident that all the features used by the machine learning model can be readily extracted from both FFPE and fresh frozen tumour cores, however, the performance of the model will need to be prospectively validated within the context of an FFPE cohort, as training was only done using a fresh frozen cohort. It is also worth noting that in modern clinical practice core frozen biopsies are increasingly being used to rapidly establish the diagnosis of malignancy and start treatment.

- Please include the rank list of features in the supplementary data to complement the color-based figures.

We agree that it is important to provide the ranked list of features and we have added a Supplementary Table (Table 6) which includes, for each feature, its overall ranking and importance zscore (illustrated in Figure 4b), as well as signed importance z-scores (illustrated in Extended Data Figure 9b).

- It would be helpful to see the performance of an ensemble model that only includes the highest ranking features shown in Figure 4b and that can be applied to FFPE tissues.

As we discussed in an earlier comment, all DNA, RNA and digital pathology features that the model uses can also be extracted from data derived from FFPE tissues, so this should not be a limitation to the application of the framework we present. As shown in Extended Data Figure 9b, all features that we have selected are used by the different classification pipelines within the model to predict response, so reducing the input feature set would most likely decrease its performance.

The generation of an FFPE model is an aim that we will work towards in the future, though this will require retraining of the same machine learning architecture that we describe in this manuscript and further validation. The benefit of the model we report is that features from any type of data modality can be added or removed to generate predictions.

D. Appropriate use of statistics and treatment of uncertainties

Please refer to comments from statistical reviewers

E. Conclusions: robustness, validity, reliability

The conclusions from the data are valid. The machine learning models were developed using a cross-validation and bootstrapping approach and individual models were fixed before testing. The ensemble approach increases the robustness. It would be helpful to understand the difference between algorithms in the outcomes prediction and to address cases that reveal a large difference in prediction results.

We are pleased to see that the referee agrees that our approach is statistically robust. We chose to include three machine learning algorithms that complement each other: random forest, as this naturally models interactions; logistic regression, as this excels at modelling smooth transitions in probabilities; and support vector classifier (SVC) as this can flexibly model large numbers of variables.

We agree that it is important to observe how the three different algorithms perform in outcome prediction. Given the space limitations in Nature we chose not to include this in the main manuscript. However, we have now generated a new extended data figure panel (Extended data Figure 9c) which shows that the output score from the three classification pipelines is highly correlated, with logistic regression having the highest correlation with SVC, and random forest having lower correlations. (Figure R2.3).

Figure R2.3 (Extended Data Figure 9b in manuscript) Correlation of the three classification pipeline scores across the training dataset (*: $p < 0.05$, **: $p < 0.01$, ***: $p < 0.001$, ****: $p < 0.0001$).

We determined cases in which the predictor score differed substantially from the model average, as suggested by the referee. We defined outliers as those in which the normalised pipeline score (pipeline score – model average score) was >2.5 standard deviations above the mean of the normalised score for the group, and for illustration highlight the top results from this analysis in Table R3.1, which confirms that by using three different machine learning algorithms, followed by model averaging, the robustness of the final predictor score increases.

	LR score	RF score	SVC score	Final model average	Response observed
Case 1	0.73	0.73	0.5	0.65	pCR
Case 2	0.72	0.34	0.5	0.52	RCB-I
Case 3	0.58	0.38	0.36	0.44	RCB-II
Case 4	0.04	0.34	0.13	0.17	RCB-III

Table R3.1 Cases with a discrepancy between pipeline predictor scores. Algorithms that give a result that is most in keeping with response observed shown in red.

We also want to highlight that our ensemble approach is based on statistical model averaging, which has been shown under certain conditions that produce estimators that minimize the mean squared error (MSE) among point estimators and with an optimal predictive distribution¹⁹ (although we note here that we haven't performed formal statistical model averaging). Given that the discrepancies within the individual classification pipeline scores lie in the differing features used by the models, which are showcased in Extended Figure 9b, we have chosen not to go through these cases individually in the manuscript.

F. Suggested improvements: experiments, data for possible revision.

Please see suggestions in section C.

G. References: appropriate credit to previous work? The reference section is adequate.

H. Clarity and context: lucidity of abstract/summary, appropriateness of abstract, introduction and conclusions

The manuscript is well written and balanced. It should be easy to understand by reader from different disciplines. The expanded data and methods sections contain enough depth to allow evaluation by subject matter experts.

We note the referee grades our manuscript as accessible to a broad audience, with experts able to delve into the finer detail of the analyses through a review of the expanded data and methods and supplementary details.

Referee 3:

This submission from Sammut et al. details the development, training, and independent validation of a multi-omic machine learning (ML) predictor of neoadjuvant therapeutic response across the major subtypes of breast cancer (ER+, HER2+, triple negative). The key finding is that the fully integrated model that incorporates clinical/pathological, DNA, RNA, digital pathology, and treatment data yielded the best performance (precision-recall AUC=0.87 in the validation set from the ARTemis trial), although the importance of treatment data was not weighted as highly by the model. Other important results include the finding that although the training set considered response as binary (pathological complete response (pCR) versus residual disease), the fully integrated model could subdivide residual disease into three categories of residual cancer burden (RCB).

Originality and significance are deemed high. ML prediction of neoadjuvant therapeutic response has long needed a rigorous incorporation of tumor microenvironment variables, including (and perhaps especially) the immune response, and to do so here through precision-recall approaches to account for imbalanced observations is a strength. An additional strength is the inclusion of an ensemble classification approach. The work is of excellent quality, results are clearly explained, the manuscript is well-written, and for the most part the study is statically robust.

We acknowledge the very positive and thorough review, including commenting on originality and significance of our study, as well as highlighting the strengths and quality of both the analysis and findings.

Three points for the authors which, if addressed, would improve understanding and perhaps uptake of this ML predictor are as follows. One, it is a little perplexing that treatment parameters/features appear to be less important to the overall integrated training model (Figure 4b, Extended Data Figure 9b), as they are omitted from the description of the results on manuscript page 12.

We agree with the referee that treatment parameters are less important to the overall training model. By including these parameters, we have shown that it is the biology of the disease that is predominantly driving response to therapy. It is worth noting that the significance is also lower due to collinearity: most HER2+ patients are treated with anthracycline followed by a taxane (+anti-HER2 therapy), whilst most HER2- patients are treated with taxane followed by

an anthracycline. As some cases were treated with anthracycline-free regimens, and there was heterogeneity in the number of chemotherapy cycles delivered, it was important to include these variables to explore whether a gain could be obtained from adding this. Extended Data Figure 9b shows that the ML algorithms do use these therapy features in the final model, showing that there is some benefit to including them.

Two, the importance and signed importance z-scores for model features in these same figures suggest that clinical/pathological data such as HER2 status, ER status, and histological grade also contribute relatively little to overall classification, although this varies between individual models.

We agree that the integrated models do not rely on clinical ER and HER2 data but rather preferentially use the expression of *ERBB2* and *ESR1* to make predictions. This has previously been reported by others, where expression was a stronger predictor of response than classical pathology phenotypes^{20,21}.

We explored this further by looking at *ESR1* and *ERBB2* expression in the discovery dataset. Figure R3.1a shows the distribution of *ERBB2* expression across all pathology HER2+ tumours. Despite all being HER2+, tumours that attained pCR or RCB-I had much higher expression of *ERBB2* than those that did not. Likewise, within clinically ER+ tumours, tumours that attained pCR had much lower *ESR1* expression (Figure R3.1b). For this reason, the machine learning framework gives a much higher weighting to the continuous RNA expression, rather than the binarized ER/HER2 pathology status (Extended Data Figure 9b). Similarly, the framework prefers to use continuous variables of proliferation (eg taxane score based on RNA expression), rather than the categorical histopathological grade.

Figure R3.1: Box plots showing the association between (a) *ERBB2* expression and response in clinically HER2+ tumours and (b) *ESR1* expression and response in clinically ER+ tumours.

Three, while the overall classifier takes an average of the three algorithms, across the board the random forest approach appears to discount immune features. What is the reason for this?

The random forest classification models naturally high-level interactions and, as shown in Extended Figure 9b, appears to be strongly and predominantly relying on lymphocyte density to model immune infiltration and using LOH HLA to a lesser degree to model immune evasion. We hypothesise that these interactions are sufficient for it to model immune modulation and, as a result, the algorithm relies less on expressed neoantigens, STAT1 score, T cell dysfunction and exclusion to make predictions.

Referee 4:

The authors collected clinical, digital pathology, genomic and transcriptomic profiles pre-treatment biopsies of breast tumors from 168 patients treated with chemotherapy +/-HER-targeted therapy prior to surgery. First, they performed statistical analysis to identify features associated with pathological complete response (pCR) using each modality separately. Then, they developed machine learning models to predict pathological complete response (pCR) using all modalities. They also validated their results using an independent dataset. The statistical and machine learning methods used seem sound. Overall, the manuscript is clear and accessible. This work would be of interest to breast cancer research community. Novelty of results are outside the scope of my expertise, and I was unable to assess fully.

We would like to thank the referee for taking the time to go through our manuscript and for the positive comments. We are particularly pleased to hear that the statistical and machine learning methods used are sound, and that the manuscript is very clear and accessible.

Specific comments

-Please define the abbreviation the first time you use it in the text (e.g., OR, CI, FDR line 135)

We have amended the manuscript and ensured that abbreviations (including OR, CI, FDR, HRD) are defined the first time they are used.

-When tested using the external cohort, models built with clinical+RNA, clinical+DNA+RNA, clinical+DNA+RNA+digital pathology and fully integrated didn't show significant performance difference. Can authors comment on this in discussion section?

Action taken: We have added the following sentence on page 12, lines 319 - 320: *The fully integrated model relied on features obtained from all modalities of data, with RNA features having the largest contribution (Fig. 4b, Extended Data Fig. 9b).*

-<https://github.com/cclab-339brca/neoadjuvant-therapy-response-predictor> -> The link was broken. I couldn't review.

The address the reviewer has provided is slightly different from the one we included in the manuscript: it seems that when the URL was copied from the manuscript the line number in the manuscript (339) was 'inserted' within the web address.

All the R code used to identify the univariate associations and generate figures is currently hosted at this github site:

<https://github.com/cclab-brca/neoadjuvant-therapy-response-predictor>

Within this main github site there is another link that points to a further repository that currently hosts the machine learning source code:

<https://github.com/micrisor/NAT-ML>

Please do let us know if you have any further issues with access. All the R code has been zipped within a password protected archive, the password to which is: **ecosystemsmalignant**

-Figures:

--All error bars and test statistics are not defined in several corresponding figure legends.

We have gone through figure legends and ensured that test statistics are appropriately defined:

1. All figures with box plots now have the following entry in their legend: *“The box bounds the interquartile range divided by the median, with the whiskers extending to a maximum of 1.5 times the interquartile range beyond the box. Outliers are shown as dots.”*
2. Panels that show data for which statistical significance has been computed using the same test have an additional grouped entry in the legend that specifies the test used.
3. For all figures which display confidence intervals, we have included the sentence: *“95% confidence intervals are shown.”* within the legend.

--Fig 4d what does dotted line represent? Marker for training is missing.

The marker for training (the dotted line) in Figure 4d is missing from the figure legend and we have now included it as shown below:

Amended Figure 4d: Marker for training included in legend.

References:

1. Symmans, W. F. *et al.* Long-Term Prognostic Risk After Neoadjuvant Chemotherapy Associated With Residual Cancer Burden and Breast Cancer Subtype. *J. Clin. Oncol.* **35**, 1049–1060 (2017).
2. Early Breast Cancer Trialists' Collaborative Group (EBCTCG). Long-term outcomes for neoadjuvant versus adjuvant chemotherapy in early breast cancer: meta-analysis of individual patient data from ten randomised trials. *Lancet. Oncol.* **19**, 27–39 (2018).
3. Rueda, O. M. *et al.* Dynamics of breast-cancer relapse reveal late-recurring ER-positive genomic subgroups. *Nature* **567**, 399–404 (2019).
4. Nielsen, T. O. *et al.* Assessment of Ki67 in Breast Cancer: Updated Recommendations From the International Ki67 in Breast Cancer Working Group. *J. Natl. Cancer Inst.* **113**, 808–819 (2021).
5. Sotiriou, C. *et al.* Gene expression profiling in breast cancer: understanding the molecular basis of histologic grade to improve prognosis. *J. Natl. Cancer Inst.* **98**, 262–72 (2006).
6. Bertucci, F. *et al.* Comparison of the prognostic value of genomic grade index, Ki67 expression and mitotic activity index in early node-positive breast cancer patients. *Ann. Oncol. Off. J. Eur. Soc. Med. Oncol.* **24**, 625–32 (2013).
7. Rouzier, R. *et al.* Nomograms to predict pathologic complete response and metastasis-free survival after preoperative chemotherapy for breast cancer. *J. Clin. Oncol.* **23**, 8331–9 (2005).
8. Lee, J. K. *et al.* Prospective comparison of clinical and genomic multivariate predictors of response to neoadjuvant chemotherapy in breast cancer. *Clin. Cancer Res.* **16**, 711–8 (2010).
9. Jin, X. *et al.* A nomogram for predicting pathological complete response in patients with human epidermal growth factor receptor 2 negative breast cancer. *BMC Cancer* **16**, 606 (2016).
10. Pu, S. *et al.* Nomogram-derived prediction of pathologic complete response (pCR) in breast cancer patients treated with neoadjuvant chemotherapy (NCT). *BMC Cancer* **20**, 1120 (2020).
11. Wishart, G. C. *et al.* PREDICT: a new UK prognostic model that predicts survival following surgery for invasive breast cancer. *Breast Cancer Res.* **12**, R1 (2010).
12. Rosenthal, R. *et al.* Neoantigen-directed immune escape in lung cancer evolution. *Nature* **567**, 479–485 (2019).
13. Newton, Y. *et al.* Large scale, robust, and accurate whole transcriptome profiling from clinical formalin-fixed paraffin-embedded samples. *Sci. Rep.* **10**, 17597 (2020).
14. Chong, I. Y. *et al.* The Mutational Concordance of Fixed Formalin Paraffin Embedded and Fresh Frozen Gastro-Oesophageal Tumours Using Whole Exome Sequencing. *J. Clin. Med.* **10**, (2021).
15. Chin, S.-F. *et al.* Shallow whole genome sequencing for robust copy number profiling of formalin-fixed paraffin-embedded breast cancers. *Exp. Mol. Pathol.* **104**, 161–169 (2018).
16. Luo, R. *et al.* Whole-exome sequencing identifies somatic mutations and intratumor heterogeneity in inflammatory breast cancer. *NPJ breast cancer* **7**, 72 (2021).

17. Ali, H. R. *et al.* Computational pathology of pre-treatment biopsies identifies lymphocyte density as a predictor of response to neoadjuvant chemotherapy in breast cancer. *Breast Cancer Res.* **18**, 21 (2016).
18. Ali, H. R. *et al.* Lymphocyte density determined by computational pathology validated as a predictor of response to neoadjuvant chemotherapy in breast cancer: secondary analysis of the ARTemis trial. *Ann. Oncol.* **28**, 1832–1835 (2017).
19. Raftery, A. E. & Zheng, Y. Discussion. *J. Am. Stat. Assoc.* **98**, 931–938 (2003).
20. Venet, D. *et al.* Copy number aberration analysis to predict response to neoadjuvant anti-HER2 therapy: results from the NeoALTTO phase III clinical trial. *Clin. Cancer Res.* (2021) doi:10.1158/1078-0432.CCR-21-1317.
21. Swain, S. M. *et al.* NSABP B-41, a Randomized Neoadjuvant Trial: Genes and Signatures Associated with Pathologic Complete Response. *Clin. Cancer Res.* **26**, 4233–4241 (2020).

Reviewer Reports on the First Revision:

Referee #1 (Remarks to the Author):

the authors have addressed my comments

Referee #2 (Remarks to the Author):

The comprehensive rebuttal to the concerns of this reviewer is appreciated. There are no further comments or concerns that need to be addressed.

Referee #3 (Remarks to the Author):

The authors have appropriately addressed my comments and questions from the first review. I have no further concerns.

Referee #4 (Remarks to the Author):

The authors provided a comprehensive revision of their work and all comments were addressed.